# Tropical cyclone storm surge probabilities for the east coast of the United States: A cyclone-based perspective

Katherine L. Towey[1], James F. Booth[1,2], Alejandra Rodriguez Enriquez[3], Thomas Wahl[3]

[1]Earth and Environmental Science, The Graduate Center, City University of New York, New York, NY, 10016, USA

[2]Earth and Atmospheric Science, The City College of New York, City University of New York, New York, NY, 10031, USA

[3]Civil, Environmental, and Construction Engineering and National Center for Integrated Coastal Research, University of Central Florida, Orlando, FL, 32816, USA

*Correspondence to*: Katherine L. Towey (ktowey@gradcenter.cuny.edu)

**Abstract.** To improve our understanding of the influence of tropical cyclones (TCs) on coastal flooding, the relationships between storm surge and TC characteristics are analyzed for 12 sites along the east coast of the United States. This analysis offers a unique perspective by first examining the relationship between the characteristics of TCs and their resulting storm surge and then determining the probabilities of storm surge associated with TCs based on exceeding certain TC characteristic thresholds. Using observational data, the statistical dependencies of storm surge on TCs are examined for these

characteristics: TC proximity, intensity, path angle, and propagation speed, by applying both exponential and linear fits to the data. At each tide gauge along the east coast of the United States, storm surge is influenced differently by these TC characteristics, with some locations more strongly influenced by TC intensity and others by TC proximity. The correlation for individual and combined TC characteristics increases when conditional sorting is applied to isolate strong TCs close to a location. The probabilities of TCs generating surge exceeding specific return levels (RLs) are then analyzed for TCs passing

within 500 km of a tide gauge, where between 6 % and 28 % of TCs were found to cause surge exceeding the 1-yr RL. If only the closest and strongest TCs are considered, the percentage of TCs that generate surge exceeding the 1-yr RL is between 30 % and 70 % at sites north of Sewell's Point, VA, and over 65 % at almost all sites south of Charleston, SC. When examining storm surge produced by TCs, single variable regression provides a good fit, while multi-variable regression improves the fit, particularly when focusing on TC proximity and intensity, which are, probabilistically, the two

most influential TC characteristics on storm surge.

## 1 Introduction

Population increases and development without adequate planning for hazards in coastal regions has led to an increase in exposure and vulnerability to coastal flooding in low-lying areas (e.g., Strauss et al., 2012; Hallegatte et al., 2013). Some of the factors that affect storm surges, which drive the largest coastal flooding events, are likely to become

worse in the future, through rising sea levels (e.g., Tebaldi et al., 2012; Sweet and Park, 2014; Moftakhari et al., 2015) and

increasing storm intensities with anthropogenic climate change (e.g., Sobel et al., 2016). The changes to these factors will influence how much destruction storm surge may cause in low-lying communities in the future (e.g., Rahmstorf, 2017), and therefore we must fully understand the relationship between surge and these factors in the current climate. The study herein will thus focus on the relationship between TC characteristics and storm surge for the east coast of the United States (US).

Along the US east coast, both tropical cyclones (TCs) and extratropical cyclones (ETCs) can create storm surges that generate major hazards to coastal areas (e.g., Zhang et al., 2000; Colle et al., 2010; Booth et al., 2016). For ETCs, different atmospheric circulation patterns can produce large surge, with the highest median surge occurring with a slow-moving ETC in conjunction with an anticyclone located to its north (Catalano and Broccoli, 2018). The most-common track paths of ETCs causing storm surge differ for the Mid-Atlantic and the Northeast US (Booth et al., 2016). Additionally, cities

that are farther north tend to have less TC-related storm surge extremes (Needham et al., 2015). This is because at higher latitudes, TCs encounter environmental conditions that do not promote the sustainability of TCs, including lower sea surface temperatures and increased wind shear associated with the jet stream, particularly later in the Atlantic hurricane season. However, even as far north as Boston, MA, four of the top 10 surge events since 1979 were caused by TCs (Booth et al., 2016).

Although both TCs and ETCs can generate surge, it is important to note that some of the energetics of the atmosphere differ for TCs and ETCs. While both TCs and ETCs are fundamentally low-pressure systems, TCs derive their energy through latent heat release over warm ocean waters, whereas ETCs gain their energy from the presence of air masses with different temperature and moisture characteristics (e.g., Jones et al., 2003; Yanase and Niino, 2015). Due to these differences in storm dynamics, flood exceedance curves for TCs and ETCs can exhibit different characteristics when

considering long timescales (i.e., 100-yr events) as more extreme events are likely to be associated with TCs (Orton et al., 2016). Thus, even though TCs occur much less frequently than ETCs along the US east coast (e.g., Booth et al., 2016), individual TCs can cause more damage as they often are associated with more moisture and stronger winds than ETCs. Therefore, it is the focus of this research to understand how differences in certain characteristics of TCs relate to storm surge.

       Several studies have utilized numerical models to assess the relationship between storm surge and TC
characteristics. Synthetic TC tracks along the Mid-Atlantic and the Northeast US have been heavily utilized to identify various relationships between surge and wind speed (Lin et al., 2010), TC tracks (Garner et al., 2017), and landfall angle (Ramos-Valle et al., 2020). Additionally, Camelo et al. (2020) simulated 21 storms in the Gulf of Mexico and along the east coast of the US and found no individual TC characteristic correlates well with storm surge. The effect of the size of hurricanes on storm surge was found to be significant in the Gulf of Mexico (e.g., Irish et al., 2008; Needham and Keim,

2014). While comparing both observed and modeled surge heights, Bloemendaal et al. (2019) affirmed that surge height is influenced by the intensity and size of TCs in addition to coastal complexities and slope. Peng et al. (2006) examined the sensitivity of surge induced by both offshore and onshore winds to wind speed and direction. Needham and Keim (2014) empirically found that storm surge correlates better with TC winds pre-landfall as opposed to winds at landfall; Roberts et al. (2015) found a similar result for all storm types. Modeling work also suggests that with anthropogenic climate change, TCs

will become stronger and peak intensity will occur at higher latitudes, and thus, changes to the intensity, frequency, and tracks of TCs are likely to impact storm surge (Knutson et al., 2020). While many studies have focused on utilizing synthetic tracks and models to better understand the relationship between storm surge and TCs, to our knowledge, no previous assessment has examined historical surge observations with a focus on surge variability relative to TC characteristics in addition to calculating storm surge exceedance probabilities based on TC characteristics. Therefore, we have designed an analysis to utilize past observations to determine the correlation between storm surge and TC characteristics as well as utilize those characteristics to determine the likelihood of surge exceeding some threshold at various locations along the eastern US.

The magnitude of storm surge at a location is also impacted by coastal characteristics, such as its bathymetry (Weaver and Slinn, 2010), wind drag coefficients and bottom friction (Akbar et al., 2017), coastal complexities (Bloemendaal et al., 2019), depth of near-shore waters combined with the astronomical tide cycle (Rego and Li, 2010; Talke et al., 2014), and geomorphic changes in the coastal regions (e.g., Familkhalili et al., 2020). While these factors are important to surge, our focus will be on characteristics related to TCs, including the TC proximity to a tide gauge, TC intensity, measured through its mean sea-level pressure (MSLP), TC path angle, and TC propagation speed, all of which can be ascertained from historical cyclone track information. Since this TC information as well as storm surge data are timestamped, we can relate the two datasets together. By utilizing this method of storm attribution, the analysis herein examines surge events and TCs in the observed record to understand empirically how TC characteristics can influence storm surge.

In this paper, we present a two-part analysis that examines how the magnitude of storm surge events associated with TCs varies based on the characteristics of the TCs themselves at various locations along the east coast of the US. Section 2 describes the data and methods used in calculating storm surge and associating storm surge events with TCs. Section 3 is divided into two parts, with part one first analyzing how TC characteristics both individually and in conjunction with one another correlate with the magnitude of storm surge. We further explore if examining TC characteristics individually or combined with one another improves the predictability of storm surge. Part two computes the return levels of storm surge and examines the likelihood of the return level of storm surge being exceeded by TCs that meet certain criteria. The paper concludes with a discussion of the results in section 4.

**2 Data and Methodology**

Section 2.1 describes how the storm surge data is calculated from the original water level data. Section 2.2 details the algorithm which associates storm surge events with TCs as well as the TC characteristics that are examined in relation to the storm surge.

## 2.1 Storm Surge Data

The water level data utilized in this analysis is obtained from the National Oceanic and Atmospheric Administration (NOAA) Tides and Currents website (NOAA, 2021). Twelve tide gauges, which record the water levels, that span along the east coast of the US were selected for this analysis (Table 1). Our analysis begins in 1946 for most sites, unless the station has data available beginning in a year later than 1946, as shown in Table 1, and ends in 2019 for all sites. It is important to note that the water level data is not continuous for all locations and thus, some sites may contain gaps in the data. The year

1946 is selected as the starting year in our analysis because in 1945, the NOAA-predicted tide and/or sea level data appeared to have a timing issue at some locations where the data were offset, which caused the difference between the sea level and the tide to have a tidal pattern.

       The water level data is initially provided in hourly time intervals. Each water level time series results from a combination of the mean sea level, astronomical tides, and non-tidal residual, which mainly contains the surge component.

While the wave setup is an important component to the water level (e.g., Phan et al., 2013; Marsooli and Lin, 2018), we neglect this component in our calculation of storm surge due to its overall complexities and its variations based on location and storm intensity. Additionally, the wave setup in the non-tidal residual is minimal because tide gauges are typically located in protected areas, such as harbors and bays. To obtain surge heights, we first remove the astronomical tide, which is provided on the NOAA Tides and Currents website (NOAA, 2021), from the water level data and then remove low-

frequency trends by subtracting a 365-day running mean of the water level for each site's water level time series. We refer to the resulting value as surge. Using hourly surge, we find the maximum surge per day and refer to this value as the maximum daily storm surge.

**Table 1. Locations of tide gauges used in analysis with their location and length of data record, which spans through 2019 for all sites. TCs are separated based on whether they undergo extratropical transition (ET TCs) or do not (non-ET TCs). The number of TCs within 500 km before and after the removal of missing MSLP values are included here. The average MSLP of TCs through the time-averaging technique within 500 km, 250 km, and 100 km, which are referenced throughout the manuscript, are included here.**

| Location | Latitude | Longitude | Starting Year of Record | Number of non-ET TCs (ET TCs) within 500 km | Number of non-ET TCs (ET TCs) within 500 km with MSLP available | Average MSLP of non-ET TCs within 500 km (hPa) | Average MSLP of non-ET TCs within 250 km (hPa) | Average MSLP of non-ET TCs within 100 km (hPa) |
|---|---|---|---|---|---|---|---|---|
| Portland, ME | 43.66°N | 70.25°W | 1945 | 34 (62) | 31 (52) | 983.5 | 987.8 | 981.1 |
| Boston, MA | 42.36°N | 71.05°W | 1945 | 50 (68) | 44 (58) | 983.1 | 983.3 | 985.0 |

| | | | | | | | | |
|---|---|---|---|---|---|---|---|---|
| Newport, RI | 41.51°N | 71.33°W | 1946 | 58 (68) | 53 (55) | 981.7 | 984.0 | 988.7 |
| New York, NY | 40.70°N | 74.02°W | 1946 | 70 (59) | 64 (51) | 984.8 | 984.5 | 984.6 |
| Sandy Hook, NJ | 40.47°N | 74.01°W | 1946 | 73 (59) | 67 (50) | 984.6 | 984.2 | 984.6 |
| Cape May, NJ | 38.97°N | 74.96°W | 1966 | 49 (36) | 49 (36) | 985.6 | 985.2 | 977.6 |
| Sewell's Point, VA | 36.95°N | 76.33°W | 1946 | 109 (52) | 101 (44) | 987.0 | 985.1 | 990.9 |
| Duck, NC | 36.18°N | 75.75°W | 1979 | 59 (28) | 59 (28) | 986.2 | 985.7 | 988.1 |
| Charleston, SC | 32.78°N | 79.92°W | 1946 | 122 (37) | 104 (31) | 986.6 | 987.1 | 988.9 |
| Fort Pulaski, GA | 32.04°N | 80.90°W | 1950 | 110 (27) | 96 (23) | 986.4 | 989.1 | 992.0 |
| Fernandina Beach, FL | 30.67°N | 81.47°W | 1946 | 128 (26) | 113 (20) | 989.5 | 988.7 | 991.4 |
| Key West, FL | 24.55°N | 81.81°W | 1950 | 113 (4) | 100 (4) | 990.7 | 985.4 | 982.0 |

## 2.2 Methods

Using our dataset of maximum daily storm surge for each site, we associate the surge events with TCs. The National Hurricane Center's Atlantic hurricane database (HURDAT2; Landsea and Franklin, 2013) is used to identify TCs. The HURDAT2 database provides at least 6-hourly observations of each TC, and in some rare instances, at a shorter time interval of 3 h; therefore, we use only the 6-hourly data for all TCs. The TC variables we utilize are its location, central MSLP minimum (units: hPa), and maximum sustained surface wind speed, defined as the maximum 1-min average wind speed at 10 m (units: knots). All TCs that pass within 500 km of a tide gauge are retained for this analysis. We initially consider a search radius of 500 km due to the typical spatial sizes of TCs, but also examine smaller search radii of 250 km and 100 km. Generally, a search radius beyond 500 km is too large when considering the spatial size of TCs (e.g., Booth et

al., 2016) as TCs located beyond 500 km from a location will have limited impacts. Distance from tide gauges to the TC centers are calculated using great circles. We then find all time steps along the TC track when the TC was within 500 km and examined what the maximum daily storm surge was at each of those time steps. We consider all TCs in the HURDAT2 database that are categorized as a tropical storm or hurricane when the storm is within 500 km, meaning their maximum sustained wind speed is at least 34 knots. Thus, if a cyclone in the database only reaches tropical depression strength during the time that it is within 500 km of a specific site, it is not included in our analysis. Additionally, we exclude any TCs that undergo extratropical transition (ET) and are classified as "extratropical" in HURDAT2 while the TC is within 500 km of a tide gauge since these TCs can no longer be considered purely tropical in nature. The percentage of TCs that undergo ET increases with latitude, with the six most northern sites in this analysis observing over 40 % of TCs that undergo ET (Table 1). Additional analysis for these six sites comparing non-ET TCs and ET TCs is presented in section 3.

To determine the maximum storm surge associated with a TC at a given location, only the time steps for when a TC was within 500 km of a tide gauge are considered as when the storm surge could be realistically attributable to a TC. First, the maximum daily storm surge that occurred on the day of each time step is assigned to each time step along the TC track. For example, if there are five time steps spaced apart by 6 h and three of the five time steps are on the same day, those three time steps would be assigned the same storm surge value – the maximum surge for that day. Then, the highest storm surge of all of these time steps within 500 km is the storm surge value attributed to a TC as it is the maximum surge produced by the TC. We note that the storm surge we find in this manner is not necessarily the storm surge that occurs at the time when the TC was closest to the tide gauge. However, if there are multiple time steps while the TC was within 500 km that have the same surge value, the closest time step along the TC track is utilized in the analysis. While it is near physically impossible for two TCs to be within 500 km of each other, the algorithm is set up such that in the case that there are multiple TCs within 500 km of a tide gauge, the closest one is the one more likely to be attributable to the storm surge and thus is the one that is retained for the analysis.

The first part of our analysis utilizes variables provided in the HURDAT2 dataset to examine how the maximum daily storm surge varies with TC proximity, intensity, path angle, and propagation speed. In our analysis of the relationships between storm surge and TC characteristics, we apply both linear and exponential fits. The residual standard error (RSE) is calculated to assess both the linear and exponential fits of each relationship where a lower RSE indicates a better fit. This method was utilized in Needham and Keim (2014) in examining the relationship between surge and wind speed. When analyzing linear fits, correlation coefficients are calculated using the Pearson method. To test for statistical significance, we use the p-value method, where we select a significance level of 5 %. The null hypothesis is that the correlation coefficient of our data sample is not significantly different from zero. If the p-value is less than the significance level of 5 %, we reject the null hypothesis and thus conclude that there is a statistically significant relationship among our data.

For TC intensity, our primary analysis uses MSLP. Since MSLP data is missing for some instances, we use the average of MSLP values that are recorded over the time window from 18 h prior to the surge maximum to 6 h post surge maximum. This choice of timing is motivated by the results of Needham and Keim (2014) who found storm surge best

correlates with TC winds 18 h prior to landfall. Additionally, we tested different time windows, shifting it forward or backward in time relative to the time of the surge maximum, including 24 h prior to 12 h post, 12 h prior to 6 h post, and 6 h prior to 6 h post and found the correlation between surge and MSLP for each time window does not vary significantly. The time window from 18 h prior to 6 h post displayed the highest correlation and was thus chosen as the time window to average TC characteristics over. Hereafter, this will be referenced with respect to other variables throughout this analysis as the time-averaging technique. If there are no recorded MSLP values during this time-averaging window, we remove the TC from our analysis. Table 1 indicates the number of TCs within 500 km for each site before and after we remove those TCs from our analysis.  We also analyzed the maximum surface wind speed as a measure of TC intensity but found that wind speed and MSLP are highly correlated (S1), and thus, we just consider MSLP as a measure of TC intensity for this analysis.

For the calculation of TC path angle, we calculate the change in latitude and longitude between time steps separated by five time steps along the track of the TC. This method allows us to examine the change in the direction of the TC over a longer period of time as opposed to between consecutive time steps. The atan2d function in MATLAB is then utilized to find the TC path angle, as this function returns the four-quadrant inverse tangent. The TC path angles range from 0° or 360° (eastward) to 90° (northward) to 180° (westward) to 270° (southward). Examples of TC tracks and their respective path angles for New York, NY and Charleston, SC are shown in figure 1. The TC path angles are not grouped relative to the site of the tide gauge, rather they are relative to the direction the TC is moving around the time of the surge maximum. For both New York and Charleston, the majority of TCs propagate toward the northeast around the time of the surge maximum, though there are many TCs that also move toward the northwest in Charleston.

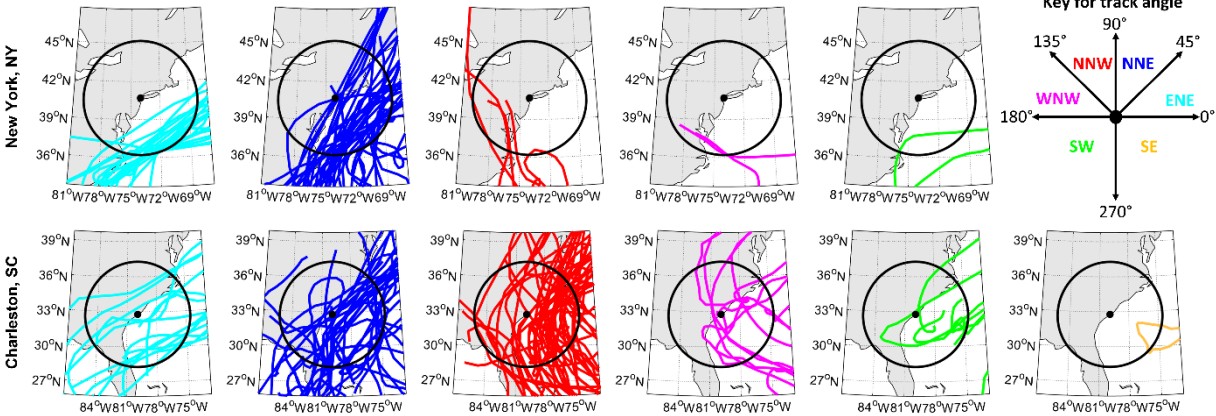

**Figure 1. Tracks of TCs within 500 km for New York, NY (top row) and Charleston, SC (bottom row) separated by path angles around the time of the surge maximum. Colors of tracks indicate the path angle as portrayed in the key and include toward the ENE (light blue, column 1), NNE (dark blue, column 2), NNW (red, column 3), WNW (magenta, column 4), SW (green, column 5), and SE (orange, column 6). The tide gauge location is indicated by the black dot and the search radius of 500 km around the location is indicated by the black circle.**

Propagation speed is calculated using the distance traveled per 6-hourly time step based on great circles. We then apply the time-averaging technique. For the data that we analyzed, however, the relationship between the surge maximum and TC propagation speed is negligible and is not included in this analysis. This does not mean that propagation speed does not have some physical impact on the surge generated by a TC, but rather that its sole influence on surge is more complex compared to the other variables that influence surge.

The second part of this analysis examines the exceedance probability of a storm surge event through calculating the storm surge heights for various return periods at each site. Surge return levels are calculated using a peaks-over-threshold method (Coles, 2001) by fitting a Generalized Pareto Distribution (GPD) to the top 1 % of daily storm surge events at each location. Before performing the fitting to the GPD, the events over the threshold are de-clustered using a 2 d window, so we satisfy the assumption of independence (e.g., Wahl et al., 2017). Return levels at 1-yr, 2-yr, 5-yr, 10-yr, and 25-yr intervals are determined from the GPD and are included in Table 3. The likelihood that a TC meets certain criteria (i.e., TC proximity of within 500 km of a location) and produces storm surge exceeding the threshold associated with a 1-yr return level is examined through a probabilistic analysis.

## 3 Results

Section 3.1 examines the correlation between storm surge and TC characteristics individually, combined, and through conditional sorting. Section 3.2 assesses the probabilities associated with TCs producing storm surge exceeding the 1-yr return level given certain TC characteristics.

### 3.1 Storm surge correlation with TC characteristics

For our correlation analysis, the first characteristic we analyze is the distance between the TC center and the tide gauge, hereafter referred to simply as TC proximity. When considering TCs that pass within 500 km of a location, the magnitude of storm surge generally increases for TCs that are closer to a given site (Fig. 2). Many of the largest storm surge events do tend to be at distances less than 200 km for most locations. However, as seen in figure 2, there are also instances where TCs close to a location generate relatively small storm surge. Conversely, there are also instances where TCs are further away from a location, but result in high storm surge (e.g., Charleston, SC in Fig. 2). For most locations, RSE is very similar when applying both linear and exponential fits, with the greatest difference seen at Newport, RI. Since we focus only on TCs that are considered purely tropical, i.e., they do not undergo ET (non-ET TCs), in this analysis, we include a supplemental figure to compare the relationship between surge and TC proximity for non-ET TCs against ET-TCs for the six most northern sites, which have at least 40% of their TCs undergo ET. When examining storm surge as a function of distance for ET TCs, the fit worsens compared to that for non-ET TCs for these six sites (S2).

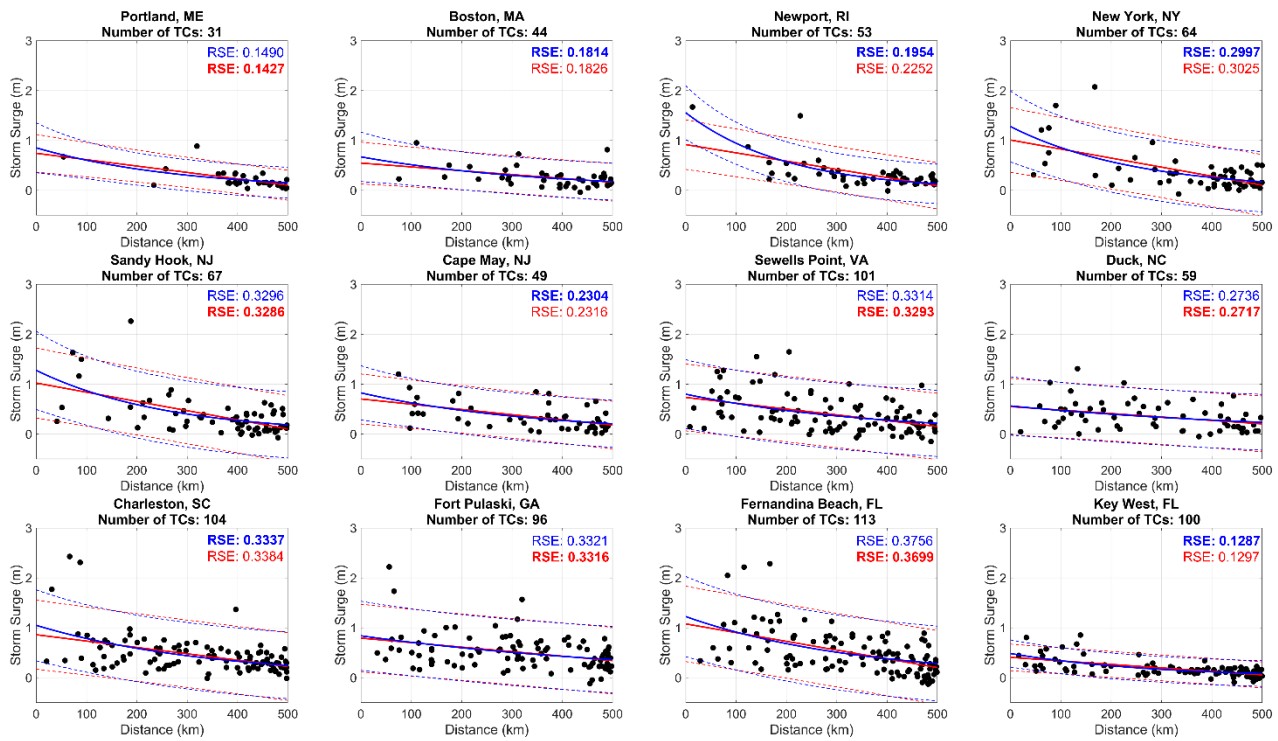

**Figure 2. Linear (red solid line) and exponential (blue solid line) fit between storm surge (m) and TC proximity (km) with 95 % confidence intervals (dashed lines) for all TCs within 500 km. Residual standard error (RSE) is provided for each type of fit with the lower value bolded.**

The second characteristic we consider is TC intensity, based on the MSLP of the TC, as discussed in Section 2.2. All locations display a similar relationship in which the magnitude of storm surge is larger for TCs with lower MSLP, which generally signifies a more intense TC (Fig. 3). Exponential fits are only shown for subsequent figures since for all figures, linear and exponential fits were found to be very similar, as was seen in figure 2. The lowest RSE is seen at Portland, ME, Boston, MA, and Key West, FL. As seen in figure 2 with TC proximity, figure 3 indicates a similar conclusion, in which TC

intensity alone does not fully explain the variability in storm surge. For some locations, such as Sandy Hook, NJ, Sewell's Point, VA, and Duck, NC, the lower RSE, as compared to the TC proximity analysis, would indicate that there is value added by examining the intensity of TCs in addition to TC proximity. We also examined the time rate of change in the MSLP of TCs and found that there was considerable variability from case-to-case and no strong statistical relationship. When examining storm surge as a function of MSLP for ET TCs, the fit worsens slightly for Portland, ME, Boston, MA, and

Newport, RI, but improves slightly for New York, NY, Sandy Hook, NJ, and Cape May, NJ (S3). This analysis of ET TCs highlights the complexities associated with the change in storm dynamics as a TC transitions into an ETC and is why we exclude these TCs from our primary analysis.

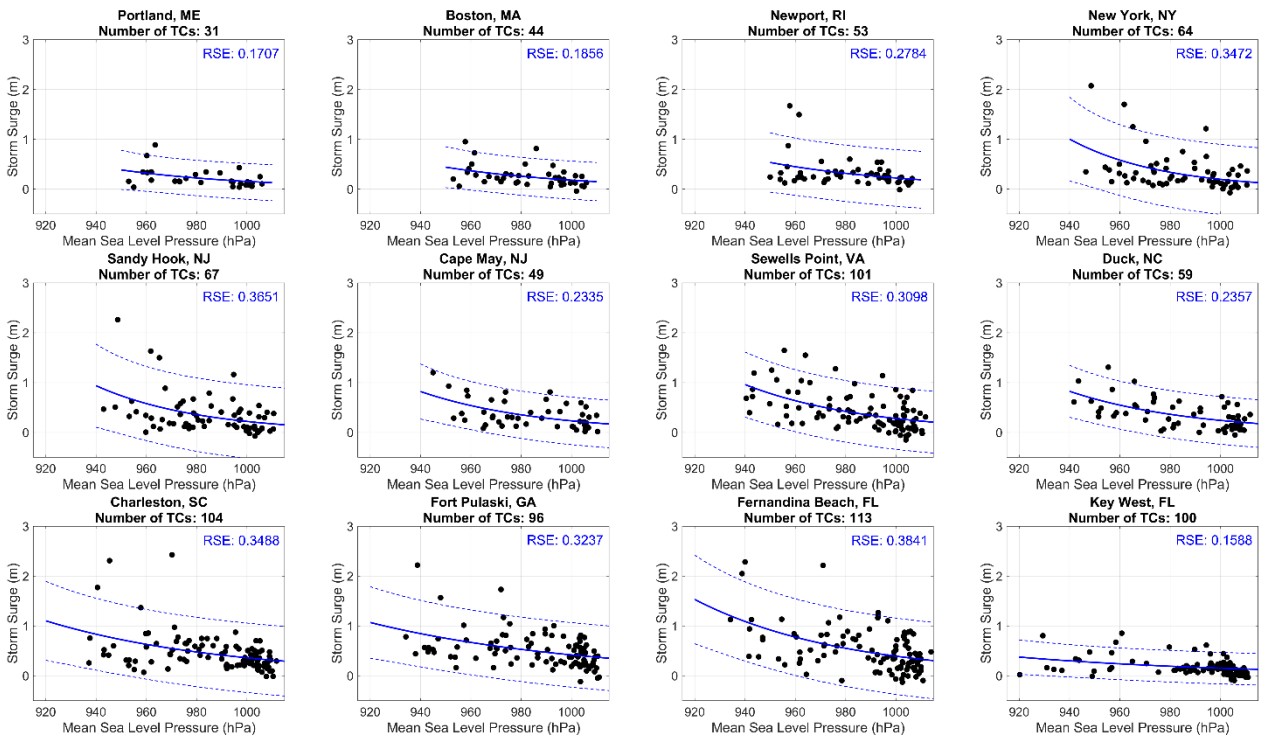

**Figure 3. Exponential fit (blue solid line) between storm surge (m) and MSLP (hPa) with 95 % confidence intervals (dashed line) for all TCs within 500 km. Residual standard error (RSE) is provided for the exponential fit.**

The path that a TC takes relative to each location is also likely to influence the magnitude of the resulting storm surge. This would be due to the direction of the onshore winds around the TC toward a tide gauge that can greatly influence storm surge. Figure 4 shows how the magnitude of storm surge varies based on the angle of the TC track relative to each location around the time of storm surge event for all TCs within 500 km. The TCs near the most northern sites along the New England coastline (e.g., Portland, ME and Boston, MA) almost exclusively move toward the northeast. For locations at lower latitudes, the range of track paths grows, with more TCs moving toward the northwest and southwest, especially for locations south of Sewell's Point, VA. For locations north of Cape May, NJ, the largest storm surge events tend to occur as TCs move toward the northeast, in which onshore winds associated with the counterclockwise flow around the TC would push water toward the coastline. Hurricane Sandy, one of the most infamous events to occur in this region, was a unique system in part to due to its southeasterly track toward the New Jersey coastline, which varied considerably from the general track direction toward the northeast that is more commonly observed in this region (Hall and Sobel, 2013). Since our primary focus in this analysis is on TCs that do not undergo ET, Sandy is not included in this analysis as it underwent ET upon approaching New Jersey. For most locations, however, there is not a significant difference in the median storm surge between different track paths (Fig. 4). The starkest difference in storm surge based on track path is seen in Fernandina Beach, FL, where TCs moving toward the east-northeast have a median storm surge of 0.23 m, whereas TCs moving toward the west-northwest have a median storm surge of 0.57 m.

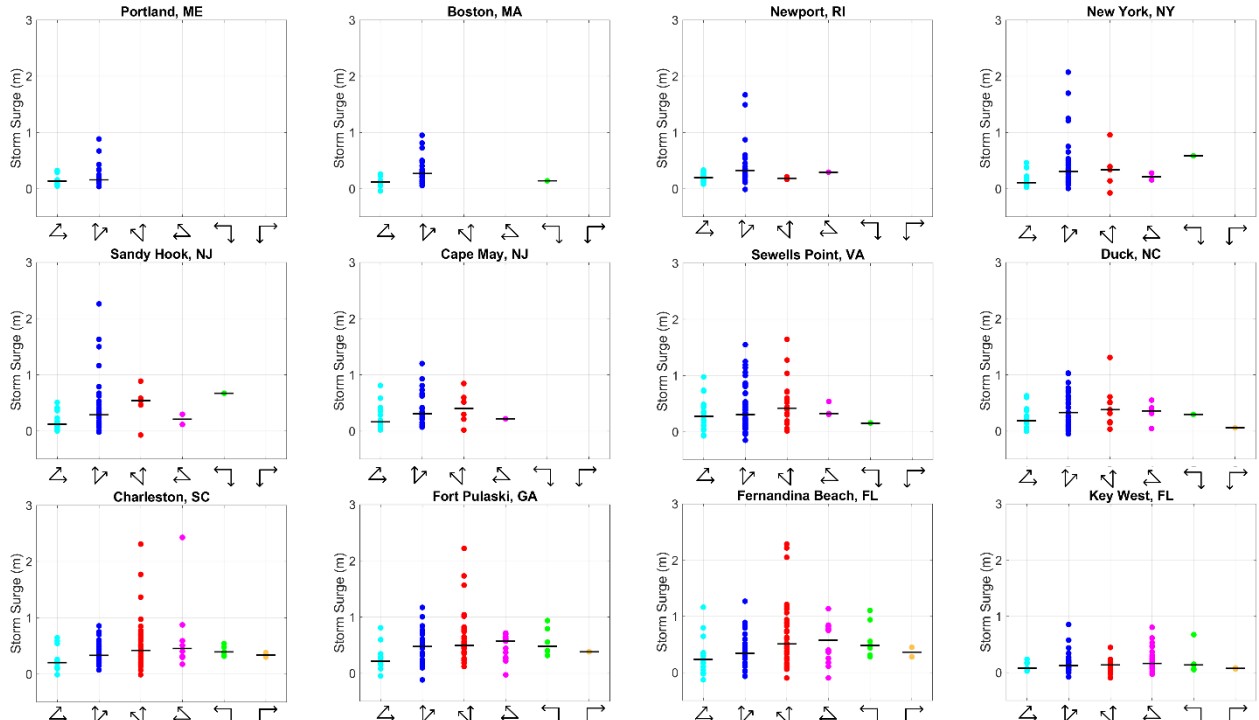

**Figure 4. Storm surge (m) separated by track path angle. Arrows along x-axis indicate range of TC track movement and is similar to track path angle key in figure 1. From left to right, arrows correspond to ENE (light blue), NNE (dark blue), NNW (red), WNW (magenta), SW (green) and SE (orange). Horizontal black line indicates the median value of storm surge for each group of track path angles.**

Individually, we have shown how the magnitude of storm surge varies based on TC proximity, intensity, and track path. We also examined the influence of propagation speed (S4) and found a negligible correlation with storm surge, suggesting that the magnitude of storm surge does not have a clear relationship with propagation speed. Next, we use conditional sorting to explore if a stronger relationship exists among these TC characteristics with storm surge.

To see how the combination of these variables can influence the predictability of storm surge, we examine how the magnitude of storm surge correlates against distance for only TCs that are stronger than the climatological average MSLP for all TCs within 500 km of a site which, hereafter are referred to as strong TCs (Fig. 5). The average MSLP is calculated for each location and is provided in Table 1. The strongest relationship is seen in Boston, MA and Key West, FL. Each data point in figure 5 is color coded based on the average track path angle for each storm surge event. For strong TCs, most locations show no discernible relationship with track path angle when analyzing storm surge and distance. For both New York, NY and Sandy Hook, NJ, which are closely located to one another, TCs that move toward the east-northeast are often associated with lower storm surge and are further away, whereas TCs that move toward the north-northeast occur at all distances and subsequently result in storm surge of both low and high magnitudes. We also used conditional sorting to examine how the magnitude of storm surge correlates with MSLP for only TCs within 250 km (S5) and saw a similar

improvement in the fit as shown in this analysis. Conditional sorting based on TC path angle and, separately, TC propagation speed, did not show a statistical relationship between TC proximity and surge.

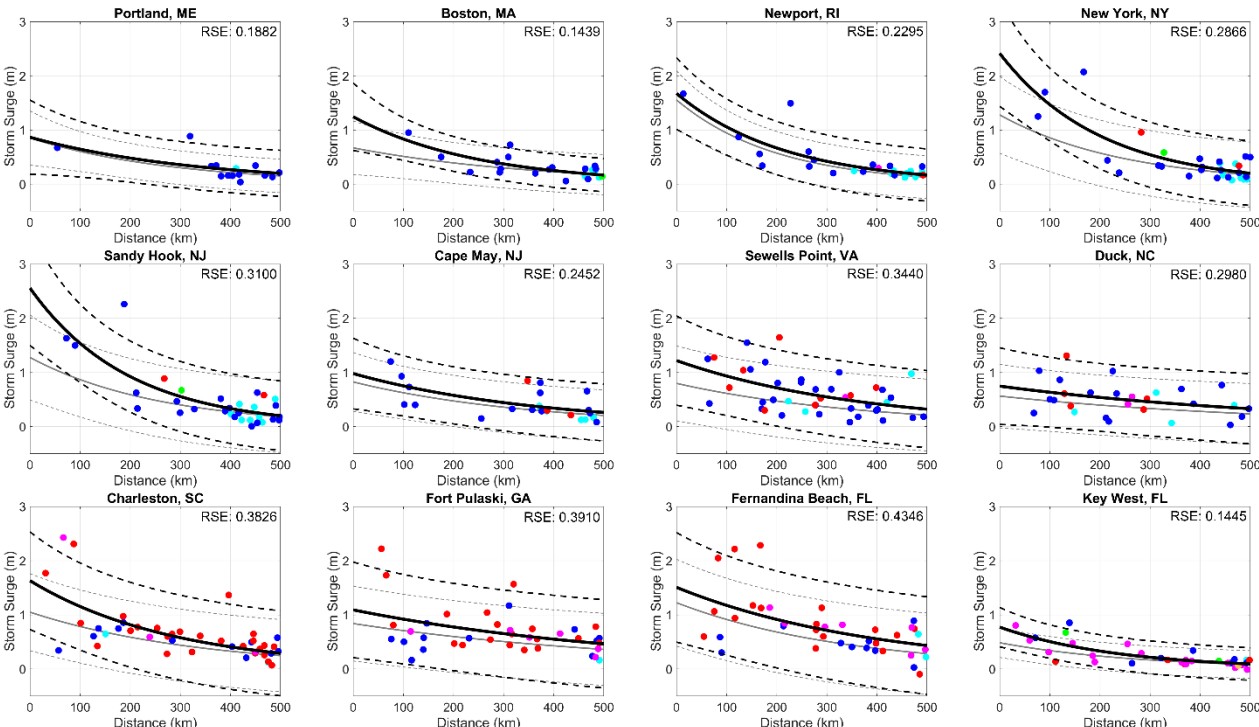

**Figure 5. Exponential fit (black solid line) between storm surge (m) and TC proximity (km) for only strong TCs within 500 km with 95 % confidence intervals (dashed black line). Exponential fit (gray solid line) and 95 % confidence intervals (dashed gray line) from Figure 2 is also included for comparison to all TCs regardless of intensity. Data points are color coded based on average track path angle as outlined in Figure 4. Residual standard error (RSE) is provided for the exponential fit.**

To complement the exponential fit analysis shown in Figures 2 - 5, we next examine correlation coefficients based on linear fits. In our linear regression analysis, we explore how the statistical fit changes if we consider multiple predictors and/or conditional sorting of the data. A comparison of the relationships between surge and TC proximity (Fig. 2) and those for surge and TC proximity after conditionally sorting to isolate for stronger TCs (Fig. 5) indicates that many locations exhibit an increase in their correlation coefficient. Table 2 displays the correlation coefficients for individual, combined, and conditionally sorted variables in their ability to predict storm surge at each location. Each location exhibits a negative correlation that is statistically significant ($p < 0.05$) at all sites based on the method described in Section 2.2 between storm surge and both TC proximity and intensity. This negative correlation suggests that as TC proximity to a location decreases, the magnitude of the storm surge increasing, highlighting the importance of TCs that are close to a location. While the relationships are statistically significant, the strength of the relationship of surge with TC proximity and intensity, individually, varies based on location. Most locations exhibit a higher correlation with TC proximity than TC intensity. Only Sewell's Point, VA, Duck, NC, and Fort Pulaski, GA exhibit a higher correlation with TC intensity than TC proximity. When TC proximity and intensity are combined as predictors of surge, the correlation increases compared to the correlation

for the variables individually and are statistically significant ($p < 0.05$) for all locations. If we isolate only TCs that are considered strong (i.e., MSLP is less than or equal to the average MSLP of all TCs within 500 km of a site) and then examine the predictability of storm surge based on TC proximity, we see that the correlation increases and is statistically significant ($p < 0.05$) for all locations except Duck, NC (Table 2, Column 5).

**Table 2. Correlation coefficients from linear analysis of storm surge with TC proximity, TC intensity, combination of TC proximity and intensity, and TC proximity for only strong TCs. Correlation coefficients that are not statistically significant have an asterisk.**

| Location | TC Proximity | TC Intensity | TC Proximity and Intensity | TC Proximity for only strong TCs |
|---|---|---|---|---|
| Portland, ME | -0.65 | -0.43 | -0.70 | -0.66 |
| Boston, MA | -0.43 | -0.41 | -0.57 | -0.70 |
| Newport, RI | -0.65 | -0.35 | -0.73 | -0.77 |
| New York, NY | -0.62 | -0.41 | -0.71 | -0.77 |
| Sandy Hook, NJ | -0.58 | -0.41 | -0.69 | -0.75 |
| Cape May, NJ | -0.52 | -0.48 | -0.66 | -0.60 |
| Sewell's Point, VA | -0.45 | -0.56 | -0.68 | -0.48 |
| Duck, NC | -0.35 | -0.58 | -0.62 | -0.34* |
| Charleston, SC | -0.46 | -0.42 | -0.63 | -0.61 |
| Fort Pulaski, GA | -0.37 | -0.43 | -0.55 | -0.39 |
| Fernandina Beach, FL | -0.54 | -0.48 | -0.68 | -0.57 |
| Key West, FL | -0.64 | -0.37 | -0.70 | -0.75 |

## 3.2 Storm surge exceedance probabilities

In considering impacts and coastal disaster planning, hazards are often ranked using return periods. These metrics

provide timescales that help in conceptualizing the potential magnitudes of the hazards, and therefore we have analyzed the

return periods for the storm surge events at our study locations. Herein we report on the relationship between the return periods and the TC characteristics using conditional sorting that builds on the lessons learned from our regression analysis in the previous section.

We calculate return levels for various return periods for each location (Table 3) using the peaks-over-threshold method as previously described in Section 2.2. Return levels are calculated using daily storm surge values for each location for all times during the year. In our analysis, we focus on the 1-yr return level, which would mean on average, a location could expect to experience one storm surge event of this magnitude each year.

Table 3. Return levels (m) for each location for return periods of 1-yr, 2-yr, 5-yr, 10-yr, and 25-yr intervals.

| Location | 1-yr | 2-yr | 5-yr | 10-yr | 25-yr |
|---|---|---|---|---|---|
| Portland, ME | 0.60 m | 0.71 m | 0.86 m | 0.97 m | 1.13 m |
| Boston, MA | 0.68 m | 0.81 m | 0.98 m | 1.11 m | 1.27 m |
| Newport, RI | 0.60 m | 0.72 m | 0.89 m | 1.04 m | 1.27 m |
| New York, NY | 0.81 m | 0.98 m | 1.22 m | 1.41 m | 1.69 m |
| Sandy Hook, NJ | 0.83 m | 1.00 m | 1.26 m | 1.47 m | 1.79 m |
| Cape May, NJ | 0.73 m | 0.85 m | 1.00 m | 1.10 m | 1.22 m |
| Sewell's Point, VA | 0.73 m | 0.88 m | 1.07 m | 1.22 m | 1.43 m |
| Duck, NC | 0.61 m | 0.71 m | 0.83 m | 0.92 m | 1.04 m |
| Charleston, SC | 0.53 m | 0.63 m | 0.80 m | 0.97 m | 1.25 m |
| Fort Pulaski, GA | 0.63 m | 0.75 m | 0.92 m | 1.06 m | 1.27 m |
| Fernandina Beach, FL | 0.76 m | 0.88 m | 1.06 m | 1.21 m | 1.45 m |
| Key West, FL | 0.20 m | 0.24 m | 0.31 m | 0.39 m | 0.55 m |

Using the 1-yr return levels, we seek to determine the probability of storm surge exceeding this threshold, conditional on certain TC characteristics (Table 4). First, we examine the probability of TCs within a specific distance resulting in storm surge exceeding the 1-yr return level. As the distance decreases from 500 km to 100 km, the percentage of TCs producing storm surge that exceeds the 1-yr return level increases. This would indicate that as a TC gets closer, the likelihood that it produces high surge is greater than if it were at a further distance. At a distance of 250 km, less than 15 %

of TCs have resulted in storm surge that exceeds the 1-yr return level at two of the three most northern sites, Boston, MA, and Newport, RI, as well as Duck, NC (Fig. 6a). Three of the four most southern sites, including Charleston, SC, Fernandina

Beach, FL, and Key West, FL, have experienced more than 30 % of TCs within 250 km resulting in storm surge exceeding the 1-yr return level, with over 50 % at Key West, FL.

**Table 4. Percentages for each location of TCs within 500 km, 250 km, and 100 km under two criteria: 1) within a specified distance that produced surge exceeding 1-yr return level and b) within a specified distance and whose MSLP is less than or equal to the average MSLP of all TCs within a specified distance that produced surge exceeding 1-yr return level. The number of individual TCs that met all criteria is given by "N" and the total number of TCs that met the distance and/or intensity criteria but did not exceed the return level is given by the bracketed number. The "N" number divided by the bracketed number will give the percentage in the same box.**

| Location | For TCs within X distance, how many produce surge exceeding 1-yr return level? | | | For strong TCs within X distance, how many produce surge exceeding 1-yr return level? | | |
|---|---|---|---|---|---|---|
| | 500 km | 250 km | 100 km | 500 km | 250 km | 100 km |
| Portland, ME | 6.45%<br>N = 2 [31] | 18.18%<br>N = 2 [11] | 50.00%<br>N = 1 [2] | 13.33%<br>N = 2 [15] | 50.00%<br>N = 2 [4] | 100.00%<br>N = 1 [1] |
| Boston, MA | 6.82%<br>N = 3 [44] | 12.50%<br>N = 2 [16] | 20%<br>N = 1 [5] | 13.04%<br>N = 3 [23] | 28.57%<br>N = 2 [7] | 50.00%<br>N = 1 [2] |
| Newport, RI | 7.55%<br>N = 4 [53] | 11.54%<br>N = 3 [26] | 25.00%<br>N = 2 [8] | 14.29%<br>N = 4 [28] | 23.08%<br>N = 3 [13] | 66.67%<br>N = 2 [3] |
| New York, NY | 7.81%<br>N = 5 [64] | 20.83%<br>N = 5 [24] | 25.00%<br>N = 3 [12] | 12.12%<br>N = 4 [33] | 36.36%<br>N = 4 [11] | 40.00%<br>N = 2 [5] |
| Sandy Hook, NJ | 7.46%<br>N = 5 [67] | 20.83%<br>N = 5 [24] | 25.00%<br>N = 3 [12] | 11.43%<br>N = 4 [35] | 36.36%<br>N = 4 [11] | 33.33%<br>N = 2 [6] |
| Cape May, NJ | 12.24%<br>N = 6 [49] | 15.79%<br>N = 3 [19] | 0.00%<br>N = 0 [4] | 20.00%<br>N = 5 [25] | 25.00%<br>N = 3 [12] | 0.00%<br>N = 0 [2] |
| Sewell's Point, VA | 15.84%<br>N = 16 [101] | 21.82%<br>N = 12 [55] | 16.67%<br>N = 3 [18] | 27.91%<br>N = 12 [43] | 34.62%<br>N = 9 [26] | 33.33%<br>N = 3 [1] |
| Duck, NC | 16.95%<br>N = 10 [59] | 14.29%<br>N = 5 [35] | 8.33%<br>N = 1 [12] | 33.33%<br>N = 9 [27] | 29.41%<br>N = 5 [17] | 20.00%<br>N = 1 [5] |
| Charleston, SC | 27.88%<br>N = 29 [104] | 33.90%<br>N = 20 [59] | 30.00%<br>N = 6 [20] | 50.00%<br>N = 22 [44] | 60.71%<br>N = 17 [28] | 71.43%<br>N = 5 [7] |
| Fort Pulaski, GA | 26.04%<br>N = 25 [96] | 27.78%<br>N = 15 [54] | 26.32%<br>N = 5 [19] | 40.00%<br>N = 16 [40] | 42.86%<br>N = 9 [21] | 42.86%<br>N = 3 [7] |
| Fernandina | 24.78% | 40.68% | 36.84% | 45.00% | 68.00% | 66.67% |

| Beach, FL | N = 28 [113] | N = 24 [59] | N = 7 [19] | N = 18 [40] | N = 17 [25] | N = 4 [6] |
|---|---|---|---|---|---|---|
| **Key West, FL** | 26.00% <br> N = 26 [100] | 52.38% <br> N = 22 [42] | 55.56% <br> N = 10 [18] | 34.38% <br> N = 11 [32] | 61.54% <br> N = 8 [13] | 71.43% <br> N = 5 [7] |


From our analysis in Section 3.1, we found that distance alone is not sufficient when considering the effect of a TC on the magnitude of storm surge. Therefore, we next report on the probability of TCs within a specific distance and with a specific intensity. Because the average TC intensity varies across our study location, instead of using a fixed intensity threshold to sort the TCs, we use the average intensity of all TCs within a specified distance per site. Herein we focus on the
TCs with MSLP lower than the site averages, i.e., the strongest $50^{th}$ percentile of TCs per site. At the smallest distance threshold analyzed, 100 km, all locations with the exception of Cape May, NJ and Duck, NC have at least a third of all TCs resulting in storm surge exceeding the 1-yr return level. Similar to before, three of the four most southern sites, including Charleston, SC, Fernandina Beach, FL, and Key West, FL have experienced more than 67 % of all TCs resulting in storm surge exceeding the 1-yr return level (Fig. 6b). In addition to these locations, however, the three most northern locations,
Portland, ME, Boston, MA, and Newport, RI experienced at least 50% of all TCs resulting in storm surge exceeding the 1-yr return level. While the number of TCs that are considered both close (< 100 km) and strong are small at these high latitudes, this analysis shows that these types of TCs at these latitudes may result in high surge if they meet this criteria.

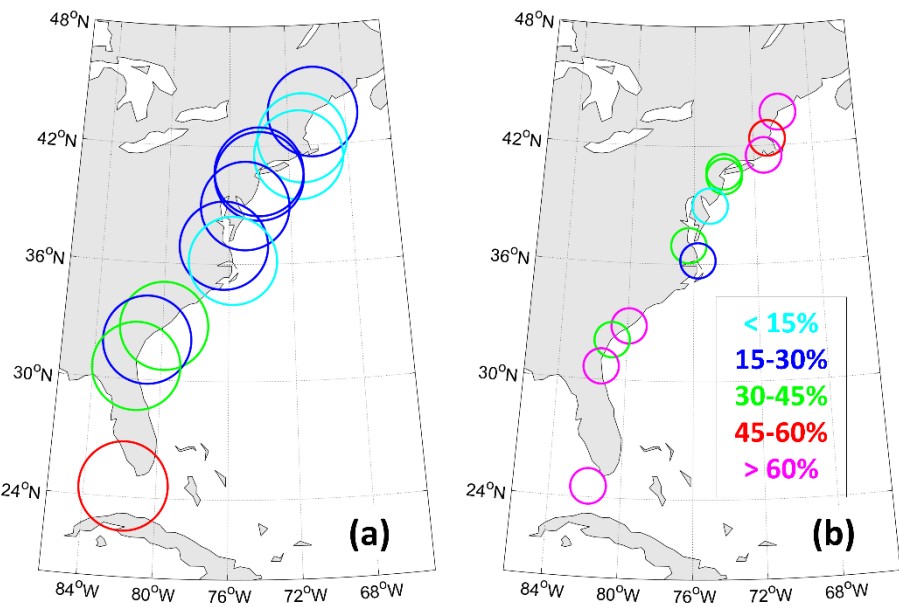

**Figure 6. Percentage of TCs that produce surge exceeding the location's 1-yr return level and: a) are within 250 km; b) are within**
**100 km and whose MSLP is less than or equal to the average MSLP of all TCs within 100 km. Size of circles indicates the search**

**radius around each location. Color coding is based on the percentage value with <15 % (light blue), 15-30 % (dark blue), 30-45 % (green), 45-60 % (red), and >60 % (magenta).**

While proximity and intensity of the TCs are important factors in predicting storm surge, we cannot ignore the role

of the TC path angle relative to each location around the time of the surge maximum. While we have shown that some locations experience TCs from a specific range of angles (Fig. 4), TC tracks with similar path angles can end up passing by a location in a different quadrant relative to the tide gauge; for example, a TC could pass to either the northwest or southeast of Charleston, SC, but have similar track path angles. In this scenario, one TC would track over land while the other TC would track over the open water. This difference could impact the structure of the TC, including its intensity and the direction of

the winds relative to the tide gauge, all of which might impact the magnitude of the storm surge. To consider this, we examine TC locations and the intensity of the TC at the time of the surge maximum (Fig. 7). For this figure, note that: (1) color now represents the strength of the TCs around the time of the surge maximum, and (2) because the surge is hourly and the TC locations are 6-hourly, the point of maximum surge for a TC corresponds to the 6-hourly time that is closest to the surge maximum.  For locations north of Sewell's Point, VA, there is a clear difference in tracks of strong TCs that do and do

not produce surge that exceeds the 1-yr return level. For TCs that do produce surge exceeding the 1-yr return level, these TCs are much stronger than the average TC and take a more meridional path whereas TCs that do not produce high surge are weaker and/or recurve out to sea. The highest surge for TCs that produce surge exceeding 1-yr return levels also generally occurs when the TC is located to the southwest of each location, allowing for onshore winds to push water towards the coastline. For locations that are further south, the picture is more complicated as TCs approach from different directions. For

these southern locations, there seems to be greater dependence on TC intensity than on TC path angle. While a majority of the TCs that produce surge exceeding the 1-yr return levels at Charleston, SC, Fort Pulaski, GA, and Fernandina Beach, FL generally move in a north-westward direction over Florida, nearly all of them have an average intensity around the time of surge maximum of 980 hPa or less.

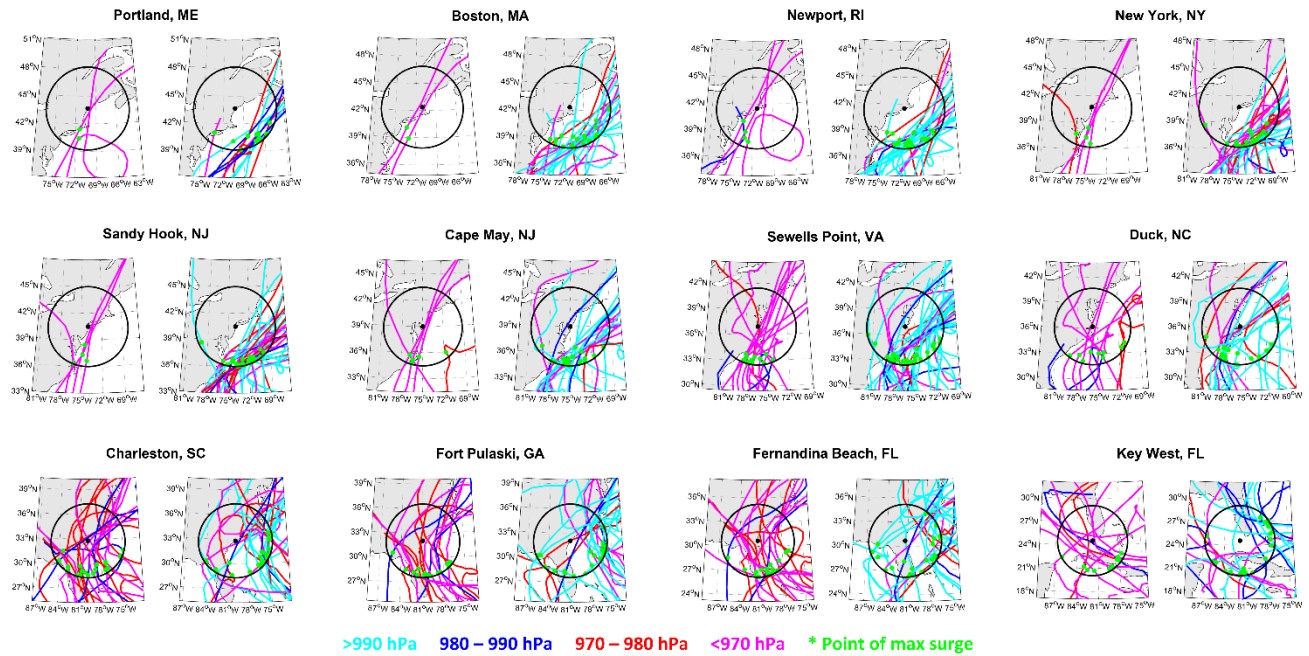

**Figure 7. For each location, strong TC tracks for those TCs that do result in surge exceeding 1-yr return level (left) and those that do not (right). Tracks are color coded based on average MSLP as follows: > 990 hPa (light blue), 980 - 990 hPa (dark blue), 970 - 980 hPa (red), and < 970 hPa (magenta). Black circle represents 500 km radius. Green dots represent closest TC location at time of surge maximum.**

## 4 Conclusion

This study uses observations to examine the predictability of storm surge based on the following TC characteristics: TC proximity, intensity, path angle, and propagation speed. At each tide gauge along the east coast of the US, storm surge is influenced differently by these TC characteristics, with some locations more strongly influenced by TC intensity (e.g., Sewell's Point, VA, Duck, NC, and Fort Pulaski, GA), but most sites more strongly influenced by TC proximity. All locations except Duck, NC see an increase in the correlation of TC proximity with storm surge once only strong TCs are considered.

When correlating storm surge with TC characteristics, we found the following for single-variable correlations: TC propagation speed does not have statistically significant relationships with surge amplitude; TC proximity and intensity both have a statistically significant ($p < 0.05$) but low to moderate correlation; TC path angle has a conditional dependence, but only at some locations. Taken together, the results indicate that storm surge produced by TCs cannot be fully explained by one TC characteristic. This result reinforces the natural variability of TCs, such that each TC is unique in its shape, size, speed, and location. Thus, it is challenging to find a strong correlation between storm surge and individual TC characteristics. For most sites, the highest storm surge occurs when a TC is within 250 km of a site and the TC intensity is

strong. This at least affirms the natural assumption that a TC that is both close to a site and strong has the greatest chance of resulting in high storm surge. Related to this point: when comparing all TCs within 500 km to those TCs considered strong within 500 km, the correlation increased for all locations except Duck, NC.

When we consider all TCs that pass within 500 km of a site, the percentage of TCs that cause surge exceeding the 1-yr return level is between 6 % and 28 %, with the higher percentages at the more southern sites. For a 100 km search radius, the percentage of TCs generating storm surge exceeding the 1-yr return level is larger at nearly all sites, with the exception of Cape May, NJ and Duck, NC, both which exhibit a decrease. If we consider only the strongest TCs, almost all sites have an increase in the probability of a 1-yr surge exceedance. Cape, May, NJ and Duck, NC are again exceptions, signifying that other factors must play an important role in storm surge generation. The site at Duck, NC is unique from the other locations because it is not near or in a bay or harbor. Meanwhile the site at Cape May, NJ is unique because it is on southern edge of a peninsula abutting the Delaware Bay. For sites that are farther south, there is a greater likelihood that TCs that pass within a fixed distance of a site will generate storm surge that exceeds the 1-yr return level. One reason for this is that TCs reach their maximum strength at lower latitudes. Another issue to consider is that for the northern sites, ETCs have a larger influence in setting the amplitude of the surge returns levels (e.g., Booth et al. 2016). With this in mind, we have started a new analysis that considers the implications of separating TCs and ETCs in probabilistic assessments.

The full complexity of the relationship between TCs and storm surge becomes apparent when we conditionally sort, based on TC intensity, the paths of TCs that do and do not generate surges that exceed the 1-yr return levels per site (Fig. 7). For some locations, there is a suggestion of a relationship with TC distance and track path angle (e.g., Newport, RI), while for other sites, the path seems less relevant than the TC intensity (e.g., Fernandina Beach, FL). Overall, the story of this analysis is three-fold: (1) using single and multi-variable regression to predict TC-generated surge in the observational record provides a good but not great fit; (2) TC proximity and intensity are better predictors than TC path angle or propagation speed, and (3) when a strong TC passes within 100 km of a location, there is always at least a 1-in-3 chance that it will generate at least an exceedance of the 1-yr return level – with two site exceptions that depend strongly on coastal geometry.

Before starting this study, we hypothesized that (based on basic physics): TC intensity would have a strong relationship with storm surge, if we were able to isolate cases in which other TC characteristics were similar. Ultimately, we found that isolating "the same type" of TC is not simple. For the southernmost sites, the relationships are more obvious, and that is possibly due to the larger sample size. For the more northern sites, one might consider testing the hypothesis using numerical modelling, in which one could model a single TC and synthetically change details of the storm, as done previously by Lin et al. (2010), Garner et al. (2017), and Ramos-Valle et al. (2020). However, we want to emphasize that such an approach is very different from our work herein, because in the observational dataset it is not possible to ensure that only one characteristic of a TC varies while all others remain constant.

While many studies have focused on the utilization of numerical models to understand the relationship between TC characteristics and storm surge, this study uses historical observations along the east coast of the US to assess the

relationship between TC characteristics and storm surge. This type of analysis allows us to understand the current relationship between TC characteristics and storm surge so that this information can be applied to the understanding of how storm surge and subsequently, the characteristics of TCs, may change under a warming climate. While no single TC characteristic determines how much surge will be generated, this analysis does offer a unique perspective on the probabilities of surge events associated with all TCs rather than only those that cause extreme surge.

This type of analysis, while limited to the east coast of the US, can be applied to any region with a record of observations associated with any type of hazard to be used in conjunction with any cyclone dataset. This cyclone-hazard association algorithm has been applied to associating precipitation and streamflow events with both TCs and ETCs in the Catskill Mountains of New York state (Towey et al., 2018). In this instance, ETC tracks were identified by applying a Lagrangian tracking algorithm (Bauer et al., 2016), which follows centers of low sea-level pressure, to reanalysis data. Similar to our analysis presented herein, Lionello et al. (2019) linked sea level anomalies to the intensity and position of cyclones in the Mediterranean Sea through the use of a cyclone tracking algorithm. Given any observational dataset for a location and a cyclone tracking algorithm, this type of analysis can be utilized to conduct similar research for any region.

**Data Availability.** Water level data that is used for the calculation of storm surge is publicly available and can be accessed at https://tidesandcurrents.noaa.gov/. Dataset of tropical cyclones is also publicly available and can be accessed at https://www.nhc.noaa.gov/data/.

**Author contribution.** KT wrote the manuscript with input from JB, ARE, and TW. JB downloaded data and calculated surge. KT completed data analysis and analyzed results with JB, ARE, and TW. JB provided code framework for cyclone association code. ARE provided code framework for calculation of return levels.

**Competing interests.** The authors declare that they have no conflict of interest.

**Acknowledgements.** This work was partially funded through NSF PREEVENTS award numbers 1854773 and 1854896.

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
