# Peer review of "Tropical cyclone storm surge probabilities for the east coast of the United States: A cyclone-based perspective"

_Natural Hazards and Earth System Sciences, 2021_

## Author Comment (AC1)

**Response to RC1 on nhess-2021-251**

Review of the paper "Tropical cyclone storm surge probabilities for the east coast of the United States: a cyclone-based perspective" by Towey et al, 2021.

Here the Authors analyze how much of surges characteristics in selected locations along the East US coast may explain by TCs producing them or crossing the areas nearby. The work is quite very interesting and absolutely worthwhile to be published. However sometimes the text and above all, the methodology are not very clear and the use of terms such as "noisy" is somehow misleading. Thus I think that text can be published after some major revisions.

Line 20-22: please reformulate

**We have rephrased this to: "*This analysis offers a unique perspective by first examining the relationship between the characteristics of TCs and their resulting storm surge and then determining the exceedance probabilities of storm surge associated with TCs based on certain TC characteristics.*"**

Lines 25-29 : I would merge these two sentences together

**We have merged these sentences together as follows: "*The conditions that generate storm surge, which drive the largest flooding events, are likely to become worse in the future, which can be attributed to rising sea levels (e.g., Tebaldi et al., 2012; Sweet and Park, 2014; Moftakhari et al., 2015), geomorphic changes in the coastal regions (e.g., Familkhalili et al., 2020), and increasing storm intensities with anthropogenic climate change (e.g., Sobel et al., 2016).*"**

Line 35 : I would avoid to use the term "scenario" because as far as I understand you are talking about "atmospheric configuration"

**Correct. We have changed "scenarios" to "atmospheric circulation patterns."**

Line 45-47 : Could you please explain what you mean with "noisy"?

**By "noisy," we are referring to variability in the data points from the statistical fit. We will replace all mentions of "noisy" in the manuscript with specific descriptions.**

Line 49: Again…what do you mean with "noisy"? how is this relationship? Linear? Please explain better

**See prior answer. This figure (6) in reference to Irish et al. (2008) does not show the type of relationship. We will reflect this change in our statement.**

Line 62 : As the sentence is formulated, it looks like that storm surges affect TC characteristics. Is it like that? Please explain.

**This sentence has been reworded to: "*Since this TC information is timestamped, storm surge data can then be related to various TC characteristics over the course of its lifecycle.*"**

Line 86-87: Could you please explain why it is not important in the statistics neglecting the wave setup? Do you have a reference for that?

**It was not our intention to suggest that it is not important to neglect the wave setup. However, for the sake of our calculations of storm surge, we do not consider the effects of the wave setup. Wave setup is very important, but also very complicated. We have rephrased this sentence to: "*While the wave setup is an important component in the water level (e.g., Phan et al., 2013, Marsooli and Lin, 2018), we neglect this component in our calculation of storm surge due to its overall complexities and its variations based on location and storm intensity.*"**

2.2 Methods section

A1) maybe I miss the point in the text but do you explain somewhere in text the choice of the radius for the search of TC in the circle centered on the location of the surge? In Figure 2 you talked about 250 km, why do not you choice 400 or 500 km?

**We will update Figure 2 to include all TCs within 500 km. We have added this sentence to the manuscript: "*All TCs that pass within 500 km of a tide gauge are retained for this analysis. We initially consider a search radius of 500 km due to the typical spatial sizes of TCs, but also examine smaller search radii of 250 km and 100 km.*"**

A2) I do not understand well the method. You consider all the TCs crossing a radius in a certain distance from the location and you associated the closest one to the event in the day of each maximum daily storm surge, do you? If so it is not very clear from text. If you have two closed systems crossing the area (probably something rare or impossible) how do you find that one responsible for the event?

**We have rephrased the relevant paragraph in this section to the following: "To determine the maximum storm surge associated with a TC at a given location, only the time steps for when a TC was within 500 km of a tide gauge are considered when the storm surge could be realistically attributable to a TC. First, the maximum daily storm surge that occurred on**

**the day of each time step is assigned to each time step. The highest storm surge of all of these time steps within 500 km is the storm surge value attributed to a TC. We note that the storm surge we find in this manner is not necessarily the storm surge that occurs at the time when the TC was closest to the tide gauge. In the case that there are multiple TCs within 500 km of a tide gauge, the closest one is the one more likely to be attributable to the storm surge and thus the TC that is retained for the analysis."**

A2) Why do you choose a liner fit as best fit and not for example an exponential fit as done in the references in the introduction?

**This is a fair point. We will apply both linear and exponential fits to our data, noting which provide the better fit for each location and variable.**

Line 154 : I think that the use of term noise is misleading. I would just state that the propagation speed is less important than other variables

**Agreed, we have changed this to: "*This does not mean that propagation speed does not have some physical impact on the surge generated by a TC, but rather that its sole influence on surge is more complex compared to the other variables that influence surge.*"**

Line 161-164 : not very clear this sentence, please rephrase

**We have rephrased this to: "*The likelihood that a TC meets certain criteria (i.e., a TC comes within 500 km of a location) and produces storm surge exceeding the threshold associated with a 1 yr return period is examined through a probabilistic analysis.*"**

Section 3.1

I would avoid to say "strong enough predictor" or "better predictor". What I see is that the variables that you consider are not able to explain completely the variability observed for the storm surges . Please reformulate

**We have changed mentions of "strong enough predictor" or "better predictor" to state that the variable does not fully explain the variability in storm surge.**

Line 215-17 : not very clear… do you mean TC stronger that the climatology of the systems crossing the area?

**Yes, we will clarify the wording in this statement to reflect this.**

Line 303-305: Not very clear..as far as I see, you correlate storm surges and TCs characteristics not the opposite. Again the adjective noisy here is not correct in the sense

that relationship between surge feature and TC, I think, is not linear not noisy. Please rephrase

**You are correct. We will rephrase this statement.**

---

## Author Comment (AC2)

**Response to RC2 on nhess-2021-251**

Review "Tropical cyclone storm surge probabilities for the east coast of the United States: A cyclone-based perspective"

General comments

In their manuscript "Tropical cyclone storm surge probabilities for the East Coast of the United States: A cyclone-based perspective", the authors seek to identify relationships between tropical cyclone (TC) characteristics and storm surge heights along the US coastline. While I believe this is a relevant topic to study, I recommend additional analyses to improve on the novelty of the research. Please find below my reasoning:

1) The authors consider the TC distance to tide gauge station, TC intensity, and TC angle at landfall in their analysis. I feel it's debatable whether the distance to a tide gauge station is something that can be truly attributed as a TC characteristics (why not consider TC size?). In addition, various past studies have (extensively) discussed similar characteristics. Many of them are already cited in the text (Lines 44 – 52) so I will not repeat them here, but these could be added for a more comprehensive overview of what's already been done:

- Needham & Keim (2014) (https://doi.org/10.1175/2013EI000558.1) who assessed the influence of storm size on hurricane surge;

- Ramos-Valle et al (2020) (https://doi.org/10.1029/2019JD031796) who extensively studied the influence of TC landfall angle on storm surges along the Mid-Atlantic Bight;

- Bloemendaal et al (2019) (https://doi.org/10.1007/s00382-018-4430-x) who also assessed the influence of various different TC and geographical characteristics on storm surges;

- Peng et al (2006) (https://doi.org/10.1016/j.ocemod.2006.03.004) on the asymmetry of storm surges and TC wind fields;

- Akbar et al (2017) (https://doi.org/10.3390/jmse5030038) on the influence of wind drag coefficients and bottom friction on Hurricane Rita's storm surge height

While I welcome research seeking additional answers to explain storm surge heights, I would strongly recommend the authors to improve on the novelty of the research to make this research truly stand out compared to the literature that's already out there.

This can be achieved through (for example) 1) including more TC and landfall (coastal slope/coastal complexity/terrain features near the tide gauge station) characteristics 2) extensively seeking for multivariate relationships 3) and to also trying different types of relationships rather than just a linear one.

**Thank you for this suggestion. Since the focus of this research is from the perspective of TCs and based on your comment about excluding TCs that underwent extratropical transition, we will include some analysis on comparing TCs that do and do not transition for select locations. Additionally, we will include an additional figure that is referenced as "not shown" in the last paragraph of the results section to highlight the distribution of TC tracks that do and do not produce surge exceeding the 1-yr return period. Lastly, we have included exponential fits in addition to linear fits of our data. We believe the inclusion of these items enhances the quality of this research.**

2) Throughout the manuscript, it seems like the authors are solely looking at TCs in their analysis. However, in the Methods-section, they say that they also include TCs that have undergone extratropical transition. These systems can no longer be considered tropical by nature (rather, extratropical), hereby having different characteristics than TCs and they should thus be excluded from the analysis.

**This is an important caveat -- thank you for bringing it to our attention. We have gone back and removed all TCs from our analysis that were classified as extratropical in HURDAT2 while the TC was within 500 km of a location.**

3) The authors use daily maximum storm surge heights and couple this with 6-hourly TC data. I don't see the added value of using daily maximum storm surge heights when the tide gauge data is provided in hourly data (see line 85) and TCs are characterized by strong spatial and temporal gradients that can strongly vary within hours.

**Since we are interested in identifying the maximum surge height produced by a TC, we believe the use of daily maximum storm surge is relevant. We are not examining aspects of surge related to the duration of surge in connection with the evolution of a TC, in which case, we agree that it would be important to utilize the original hourly surge data due to the strong spatial and temporal gradients observed in TCs. However, we are identifying the highest storm surge produced per day (while retaining the hour at which this occurred) and then associating that with the nearest 6-hourly TC observation. For the purpose of our research, this method allows us to examine the TC characteristics around the time the surge maximum occurred at a location.**

4) The results-section could benefit from some in-depth discussion of why the spatial differences emerge in relation to typical TC behavior/patterns.

**We will include further discussion on this topic.**

Introduction – specific comments

Nowhere in the introduction is there any mention of the TC characteristics that will be under consideration in this manuscript. Please add this description.

**We do mention these characteristics in the second to last paragraph of the introduction. However, we will be even more specific in the characteristics under consideration and add that in the introduction.**

The introduction (more specifically, the second paragraph, lines 33 – 44) makes it seem like there will also be a focus on ETCs. Please make it explicitly clear you will solely focus on the TCs.

**We will explicitly state that ETCs are not the focus of this research.**

Line 24-25: exposure is not the same as vulnerability

**This is an important differentiation and we will address the usage of these terms.**

Line 29-30: The amount of destruction is also influenced by changes in exposure and vulnerability

**We will add this into our introduction.**

Line 30-31: What do you mean with this sentence?

**We mean that the relationship between surge and atmospheric/oceanic variables may not be linear. We will replace the use of the word "monotonic" with this description.**

Line 35: Could you please elaborate on what these differences exactly are?

**This sentence was updated to: "For ETCs, different atmospheric circulation patterns can produce large surge, with the highest median surge occurring with a slow-moving ETC in conjunction with an anticyclone located to its north (Catalano and Broccoli, 2018)."**

Line 36 – 38: Please explain to the reader why these cities have less TC-related storm surge extremes (along the lines of ocean waters are colder + more wind shear, so less favorable for TCs), that will also help the reader understand why this Boston example is noteworthy.

**We have included more information as: "This is because at higher latitudes, TCs encounter unfavorable environmental conditions that promote sustainability of TCs, including cooler sea surface temperatures and increased wind shear associated with the fluctuating jet stream, particularly later in the Atlantic hurricane season."**

Line 38 – 44: Please fill in the gaps that are left in this paragraph: 1) what are the differences in storm dynamics? 2) What are the different characteristics of the flood exceedance curves? 3) What exactly is the frequency of TCs compared to ETCs?  4) How can they cause more damage?

**We will address all of these important topics and include them in this paragraph.**

Line 45 – 54: Could you please summarize this in a few sentences? Also: the term "noisy" is very vague. I also feel like the literature is very tailored towards US case studies and misses some other relevant studies (see my earlier comment)

**We will summarize these statements and include the other relevant studies suggested earlier. Additionally, we have removed the use of the term "noisy" and include better descriptions of the data where applicable. Since our research focuses on surge along the east coast of the US, this literature is most relevant in describing the motivation for our research.**

Line 54 – 56: I strongly disagree with the wording here. The authors make it seem like they will overcome the regional scale, but they still perform a regional assessment (namely, the US East Coast).

**Our suggestion was to imply that not many studies have used past observations and connected them to TC characteristics. Many of the studies cited here have adjusted TC tracks or used model simulations as opposed to what we are doing in utilizing past observations, isolating the maximum surge produced by a TC and examining what characteristics those TCs exhibited at the time of the surge maximum. We will rephrase this section to describe this aspect of our research in addition to our exploration of storm surge exceedance probabilities.**

Line 59 – 60: Perhaps good to also mention coastal complexity here (Bloemendaal et al 2019)

**We will include this.**

Methods – specific comments

Line 86 – 87: This is quite a bold statement to make, without any additional clarification. How big is the contribution of TC waves to total water levels?

**We have reworded this sentence to account for the complexities associated with the wave setup and its contribution to surge: "While the wave setup is an important component in the water level (e.g., Phan et al., 2013, Marsooli and Lin, 2018), we neglect this component in our calculation of storm surge due to its overall complexities and its variations based on location and storm intensity."**

Table 1: Please round the pressure to one decimal place

**We will make this change.**

Line 104: What do you mean with wind intensity? Wind speed? What is the time reference for this wind speed? (1-min, 10-min, 3-sec?) Please also add units with every TC characteristic listed here.

**Wind intensity is the maximum sustained surface wind speed, as defined in the HURDAT2 database. We have added this information to the methods.**

Line 105: Please state the exact dimensions of the "specified distance"

**We have clarified this.**

Line 123: Why are you solely testing for linear relationships?

**We expanded this to other fits and will now include information about both linear and exponential fits.**

Line 130: Please explain to the reader what these results are

**We have included this now.**

Figure 2: Why are you differentiating between a radius of 250 km (in Figure 2) and 500 km (in the text)? Also, how did you derive the track angle? To me it seems like one of the

green tracks for Charleston has a N-NNW angle upon approaching the landfall location, but it is listed as SW.

**Figure 2 has been updated to be for a radius of 500 km. This is described in paragraph 5 of section 2. The calculated track angle is not relative to landfall location, it is relative to the time of the surge maximum, so depending on when the surge maximum occurs, the TC can be moving in various directions. Additionally, we average the track angle over the time period from 18 hours prior to the surge maximum to 6 hours post surge maximum. We will add further clarification to this section regarding the calculation.**

Results – specific comments

To me, a 0.5-yr return level of 0.8m seems like a lot. This implies that (assuming the authors correctly identified individual storm surge events) Sandy Hook is affected by TC storm surges of this height on average twice per year! Could you please validate these results against other studies?

**Thanks for alerting us of this. We found in our calculation of return periods that we were using hourly data and the independence threshold was in hours not days, so this obviously skewed our results. We have now updated this for daily data and include an independence threshold of 2 days and the results make more sense now.**

Please quantify the statistical significance and correlation throughout the results-section.

**We will address this in the results section.**

A lot of results aren't shown (indicated by "not shown" in the text). Could you please add these results to the supplementary materials, so that the reader can have a look at these results?

**We will include these figures in the supplemental material.**

The continuous switching between a 500 km and 250 km radius is highly confusing – please re-read this section carefully and try to homogenize this usage of radii.

**The point of using these different radii is to illustrate the importance of distance with respect to surge, with closer (and stronger) TCs more frequently associated with higher surge. We will clarify any confusion in this section.**

Conclusions

Line 329 – 333: Please check the work of Ramos-Valle et al (2020); they synthetically changed details of various storms in the Mid-Atlantic Bight.

**We have included this citation now in our discussion.**

---

## Author Response (AR1)

Response to the editor and reviewers:

Thank you for taking the time to review this manuscript and provide constructive feedback. We have addressed the concerns of both reviewers and believe these changes have helped improve the clarity and impact of our results. A few of the changes we made were significant enough that we wanted to summarize them here:

- Reviewer #2 suggested that we separate out the TCs that underwent extratropical transition (ET) in our analysis, since these storms were not purely tropical. We agree and have removed them in the revised analysis in order to focus on purely tropical TCs. We include a comparison of some ET and non ET events for the 6 northern sites as supplemental figures.
- Additionally, reviewer #2 brought to our attention the high return level for Sandy Hook, which prompted us to check our code. We discovered we had used hourly data and the independence threshold implemented was in hours not days. So we went back and used daily data instead, which was our original intention, and this changed our return levels to more reasonable numbers.
- Lastly, to improve the overall clarity of the manuscript by highlighting the unique aspect of our work and to emphasize new findings, the introduction and methods have been updated.

**Starting on the following page, we provide point-by-point responses to all questions and suggestions from the reviewers.**
- The reviewer comments are in black.
- **Our responses are indented and written in a bold blue.**
- **The line numbers in the revised manuscript that correspond to the changes made are added to the end of each comment as [L#-#].**

Review of the paper "Tropical cyclone storm surge probabilities for the east coast of the United States: a cyclone-based perspective" by Towey et al, 2021.

Here the Authors analyze how much of surges characteristics in selected locations along the East US coast may explain by TCs producing them or crossing the areas nearby. The work is quite very interesting and absolutely worthwhile to be published. However sometimes the text and above all, the methodology are not very clear and the use of terms such as "noisy" is somehow misleading. Thus I think that text can be published after some major revisions.

Line 20-22: please reformulate

**We have rephrased this to:**

*"This analysis offers a unique perspective by first examining the relationship between the characteristics of TCs and their resulting storm surge and then determining the probabilities of storm surge associated with TCs based on exceeding certain TC characteristic thresholds." [L10 – 13]*

Lines 25-29 : I would merge these two sentences together

**We have merged these sentences together as follows:**

*"Some of the factors that affect storm surges, which drive the largest coastal flooding events, are likely to become worse in the future, through rising sea levels (e.g., Tebaldi et al., 2012; Sweet and Park, 2014; Moftakhari et al., 2015) and increasing storm intensities with anthropogenic climate change (e.g., Sobel et al., 2016)." [L27 – 30]*

Line 35 : I would avoid to use the term "scenario" because as far as I understand you are talking about "atmospheric configuration"

**Good point, thank you. We have changed "scenarios" to "atmospheric circulation patterns." [L36]**

Line 45-47 : Could you please explain what you mean with "noisy"?

**By "noisy," we are referring to variability in the data points from the statistical fit. We have replaced all mentions of "noisy" in the manuscript with specific descriptions of the data when applicable.**

Line 49: Again…what do you mean with "noisy"? how is this relationship? Linear? Please explain better

**By "noisy," we are referring to variability in the data points from the statistical fit. We removed mention of the relationship here as Irish et al. (2008) does not explicitly state the type of statistical relationship.**

Line 62 : As the sentence is formulated, it looks like that storm surges affect TC characteristics. Is it like that? Please explain.

**TC characteristics influence storm surge. We have removed this sentence and replaced it with this sentence, which better connects the preceding and following sentence:**

*"Since this TC information as well as storm surge data are timestamped, we can relate the two datasets together." [L76 – 77]*

Line 86-87: Could you please explain why it is not important in the statistics neglecting the wave setup? Do you have a reference for that?

**It was not our intention to suggest that the effect of the wave setup is something that should be neglected. However, for the sake of our calculations of storm surge, we do not consider the effects of the wave setup. We have summarized our reasoning behind this in the following statement, which has been added to the text:**

*"While the wave setup is an important component to the water level (e.g., Phan et al., 2013, Marsooli and Lin, 2018), we neglect this component in our calculation of storm surge due to its overall complexities and its variations based on location and storm intensity. Additionally, the wave setup in the non-tidal residual is minimal because tide gauges are typically located in protected areas, such as harbors and bays." [L102 – 105]*

2.2 Methods section

A1) maybe I miss the point in the text but do you explain somewhere in text the choice of the radius for the search of TC in the circle centered on the location of the surge? In Figure 2 you talked about 250 km, why do not you choice 400 or 500 km?

**We have updated Figure 2 to reflect TCs within 500 km as opposed to 250 km. Additionally, we have added this sentence to the manuscript to explain the reasoning for search radii:**

*"We initially consider a search radius of 500 km due to the typical spatial sizes of TCs, but also examine smaller search radii of 250 km and 100 km. Generally, a search radius beyond 500 km is too large when considering the spatial size of TCs (e.g., Booth et al., 2016)." [L121 – 123]*

A2) I do not understand well the method. You consider all the TCs crossing a radius in a certain distance from the location and you associated the closest one to the event in the day of each maximum daily storm surge, do you? If so it is not very clear from text. If you have two

closed systems crossing the area (probably something rare or impossible) how do you find that one responsible for the event?

**To clarify the method, we have rephrased the relevant paragraph in this section to the following:**

*"To determine the maximum storm surge associated with a TC at a given location, only the time steps for when a TC was within 500 km of a tide gauge are considered as when the storm surge could be realistically attributable to a TC. First, the maximum daily storm surge that occurred on the day of each time step is assigned to each time step along the TC track. For example, if there are five time steps spaced apart by 6 h and three of the five time steps are on the same day, those three time steps would be assigned the same storm surge value – the maximum surge for that day. Then, the highest storm surge of all of these time steps within 500 km is the storm surge value attributed to a TC as it is the maximum surge produced by the TC. We note that the storm surge we find in this manner is not necessarily the storm surge that occurs at the time when the TC was closest to the tide gauge. However, if there are multiple time steps while the TC was within 500 km that have the same surge value, the closest time step along the TC track is utilized in the analysis. While it is near physically impossible for two TCs to be within 500 km of each other, the algorithm is set up such that in the case that there are multiple TCs (or ETCs in future analyses) within 500 km of a tide gauge, the closest one is the one more likely to be attributable to the storm surge and thus is the one that is retained for the analysis."* **[L133 – 144]**

A2) Why do you choose a liner fit as best fit and not for example an exponential fit as done in the references in the introduction?

**This is a fair point. In the revised manuscript we have applied both linear and exponential fits to our data, noting which provide the better fit for each location and variable.**

Line 154 : I think that the use of term noise is misleading. I would just state that the propagation speed is less important than other variables

**Agreed, we have changed this to:**

*"This does not mean that propagation speed does not have some physical impact on the surge generated by a TC, but rather that its sole influence on surge is more complex compared to the other variables that influence surge."* **[L183 – 185]**

Line 161-164 : not very clear this sentence, please rephrase

**We have rephrased this to:**

*"The likelihood that a TC meets certain criteria (i.e., TC proximity of within 500 km of a location) and produces storm surge exceeding the threshold associated with a 1-yr return level is examined through a probabilistic analysis."* **[L191 – 193]**

Section 3.1

I would avoid to say "strong enough predictor" or "better predictor". What I see is that the variables that you consider are not able to explain completely the variability observed for the storm surges . Please reformulate

**This is another good point, thank you for noting it. We have changed mentions of "strong enough predictor" or "better predictor" to state that the variable does not fully explain the variability in storm surge.**

Line 215-17 : not very clear… do you mean TC stronger that the climatology of the systems crossing the area?

**Yes, we have clarified the wording in this statement to reflect this:**

*"To see how the combination of these variables can influence the predictability of storm surge, we examine how the magnitude of storm surge correlates against distance for only TCs that are stronger than the climatological average MSLP for all TCs within 500 km of a site which, hereafter are referred to as strong TCs (Fig. 5)." [L249 – 251]*

Line 303-305: Not very clear..as far as I see, you correlate storm surges and TCs characteristics not the opposite. Again the adjective noisy here is not correct in the sense that relationship between surge feature and TC, I think, is not linear not noisy. Please rephrase

**We have rephrased this sentence to the following:**

*"When correlating storm surge with TC characteristics, we found the following for single-variable correlations: TC propagation speed does not have statistically significant relationships with surge amplitude; TC proximity and intensity both have a statistically significant ($p < 0.05$) but low to moderate correlation; TC path angle has a conditional dependence, but only at some locations. Taken together, the results indicate that storm surge produced by TCs cannot be fully explained by one TC characteristic." [L357 – 361]*

**Response to RC2 on nhess-2021-251**

Review "Tropical cyclone storm surge probabilities for the east coast of the United States: A cyclone-based perspective"

General comments

In their manuscript "Tropical cyclone storm surge probabilities for the East Coast of the United States: A cyclone-based perspective", the authors seek to identify relationships between tropical cyclone (TC) characteristics and storm surge heights along the US coastline. While I believe this is a relevant topic to study, I recommend additional analyses to improve on the novelty of the research. Please find below my reasoning:

1) The authors consider the TC distance to tide gauge station, TC intensity, and TC angle at landfall in their analysis. I feel it's debatable whether the distance to a tide gauge station is something that can be truly attributed as a TC characteristics (why not consider TC size?). In addition, various past studies have (extensively) discussed similar characteristics. Many of them are already cited in the text (Lines 44 – 52) so I will not repeat them here, but these could be added for a more comprehensive overview of what's already been done:

- Needham & Keim (2014) (https://doi.org/10.1175/2013EI000558.1) who assessed the influence of storm size on hurricane surge;

- Ramos-Valle et al (2020) (https://doi.org/10.1029/2019JD031796) who extensively studied the influence of TC landfall angle on storm surges along the Mid-Atlantic Bight;

- Bloemendaal et al (2019) (https://doi.org/10.1007/s00382-018-4430-x) who also assessed the influence of various different TC and geographical characteristics on storm surges;

- Peng et al (2006) (https://doi.org/10.1016/j.ocemod.2006.03.004) on the asymmetry of storm surges and TC wind fields;

- Akbar et al (2017) (https://doi.org/10.3390/jmse5030038) on the influence of wind drag coefficients and bottom friction on Hurricane Rita's storm surge height

While I welcome research seeking additional answers to explain storm surge heights, I would strongly recommend the authors to improve on the novelty of the research to make this research truly stand out compared to the literature that's already out there. This can be achieved through (for example) 1) including more TC and landfall (coastal slope/coastal complexity/terrain features near the tide gauge station) characteristics 2) extensively seeking for multivariate relationships 3) and to also trying different types of relationships rather than just a linear one.

**Thank you for this suggestion and the references. We have now included mention of these in our introduction and our conclusion sections.**

We see your perspective and have added additional analysis to expand our work, as you suggested. These additions include the following:

1. We have included tests for exponential relationships in addition to linear fits of our data. We believe the inclusion of these items enhances the quality and novelty of this research.
2. Based on your subsequent comment about excluding TCs that underwent extratropical transition (ET), we included supplemental figures for select locations comparing the relationship between surge and TC characteristics for TCs that did and did not undergo ET as additional analysis.
3. We have added a new figure (Fig. 7) to highlight the distribution of TC tracks, based on TC intensity, for cyclones that did and did not produce surge exceeding the 1-yr return level. This additional analysis merges information from the probabilistic analysis and the statistical TC characteristics analysis, and we think it will be very useful for the surge and TC hazards community.

These new components have improved the scope and story of our analysis, so thank you for the suggestion.

These new elements can be combined with the fact that our work: (i) focuses on a larger and different expanse than previous work; (ii) uses a longer temporal length of the data than previous work, (iii) utilizes a method of comparing statistical relationships after conditionally sorting the data, to make a convincing case that we are providing research that will truly stand out.

We also want to address your suggestion at the onset about examining TC size. We agree that this would be a nice component. However, to our knowledge, consistent TC size estimates for the Atlantic basin are only possible in the satellite era, maybe even only from 1988 onward (e.g., Chavas et al., 2016; Wang and Toumi, 2017). Since we were interested in looking at larger datasets to boost our statistical robustness, we examined data back to 1946. Thus, for our analysis purposes, we feel that the distance of the TC to the gauge is a reasonable characteristic to utilize.

You also made some good suggestions about the possibility of examining the details of the coastlines. This is a nice idea. However, since the focus of our research is from "a cyclone perspective" and the characteristics of TCs, we choose not to focus on the characteristics related to the coastline, which we note in the introduction are important to consider, but do not fall under the scope of our research objective.

Chavas, D. R., Lin, N., Dong, W., and Lin, Y.: Observed tropical cyclone size revisited, J. Clim., 29, 2923 – 2939, doi:10.1175/JCLI-D-15-0731.1, 2016.

Wang, S. and Toumi, R.: A historical analysis of the mature stage of tropical cyclones, Int. J. Climatol., 38, 2490 – 2505, doi:10.1002/joc.5374, 2017.

2) Throughout the manuscript, it seems like the authors are solely looking at TCs in their analysis. However, in the Methods-section, they say that they also include TCs that have undergone extratropical transition. These systems can no longer be considered tropical by nature (rather, extratropical), hereby having different characteristics than TCs and they should thus be excluded from the analysis.

**This is an important caveat -- thank you for bringing it to our attention. We have gone back and removed all TCs from our analysis that were classified as extratropical in HURDAT2 while the TC was within 500 km of a location. This is reflected in these additional sentences which have been added to the manuscript:**

> *"Additionally, we exclude any TCs that undergo extratropical transition (ET) and are classified as "extratropical" in HURDAT2 while the TC is within 500 km of a tide gauge since these TCs can no longer be considered purely tropical in nature. The percentage of TCs that undergo ET increases with latitude, with the six most northern sites in this analysis observing over 40 % of TCs that undergo ET (Table 1). Additional analysis for these six sites comparing non-ET TCs and ET TCs is presented in section 3." [L128 – 132]*

**Additionally, we included supplemental figures (S2 and S3) for the six most northern locations (these sites have > 40 % of TCs within 500 km undergoin ET) where we compare the statistical relationship between surge and TC characteristics for TCs that did and did not undergo ET.**

3) The authors use daily maximum storm surge heights and couple this with 6-hourly TC data. I don't see the added value of using daily maximum storm surge heights when the tide gauge data is provided in hourly data (see line 85) and TCs are characterized by strong spatial and temporal gradients that can strongly vary within hours.

**Thanks for this comment. It helped us realize that we needed to improve our description of our methods. In brief: we start from the hourly data, but then identify the daily maxima and associate those with the TC for the 24-hour period if the TC is within our chosen search radius of the gauge. For the purpose of our research, this method allows us to examine the TC characteristics around the time the surge maximum occurred at a location. We have clarified this algorithm in the manuscript as follows:**

> *"To determine the maximum storm surge associated with a TC at a given location, only the time steps for when a TC was within 500 km of a tide gauge are considered as when the storm surge could be realistically attributable to a TC. First, the maximum daily storm surge that occurred on the day of each time step is assigned to each time step along the TC track. For example, if there are five time steps spaced apart by 6 h and three of the five time steps are on the same day, those three time steps would be assigned the same storm surge value – the maximum surge for that day. Then, the highest storm surge of all of these time steps within 500 km is the storm surge value attributed to a TC as it is the maximum surge produced by the TC. We note that the storm surge we find in this manner is not necessarily the storm surge that occurs at the*

*time when the TC was closest to the tide gauge. However, if there are multiple time steps while the TC was within 500 km that have the same surge value, the closest time step along the TC track is utilized in the analysis. While it is near physically impossible for two TCs to be within 500 km of each other, the algorithm is set up such that in the case that there are multiple TCs (or ETCs in future analyses) within 500 km of a tide gauge, the closest one is the one more likely to be attributable to the storm surge and thus is the one that is retained for the analysis."* **[L133 – 144]**

**We also note, since we are interested in identifying the maximum surge height produced by a TC, we believe the use of daily maximum storm surge is reasonable. We are not examining aspects of surge related to the duration of surge in connection with the evolution of a TC, in which case, we agree that it would be important to utilize the original hourly surge data due to the strong spatial and temporal gradients observed in TCs. However, we are identifying the highest storm surge produced per day (while retaining the hour at which this occurred) and then associating that with the nearest 6-hourly TC observation.**

4) The results-section could benefit from some in-depth discussion of why the spatial differences emerge in relation to typical TC behavior/patterns.

**We have added this detail with the inclusion of figure 7 as follows:**

*"While proximity and intensity of the TCs are important factors in predicting storm surge, we cannot ignore the role of the TC path angle relative to each location around the time of the surge maximum. While we have shown that some locations experience TCs from a specific range of angles (Fig. 4), TC tracks with similar path angles can end up passing by a location in a different quadrant relative to the tide gauge; for example, a TC could pass to either the northwest or southeast of Charleston, SC, but have similar track path angles. In this scenario, one TC would track over land while the other TC would track over the open water. This difference could impact the structure of the TC, including its intensity and the direction of the winds relative to the tide gauge, all of which might impact the magnitude of the storm surge. To consider this, we examine TC locations and the intensity of the TC at the time of the surge maximum (Fig. 7). For this figure, note that: (1) color now represents the strength of the TCs around the time of the surge maximum, and (2) because the surge is hourly and the TC locations are 6-hourly, the point of maximum surge for a TC corresponds to the 6-hourly time that is closest to the surge maximum. For locations north of Sewell's Point, VA, there is a clear difference in tracks of strong TCs that do and do not produce surge that exceeds the 1-yr return level. For TCs that do produce surge exceeding the 1-yr return level, these TCs are much stronger than the average TC and take a more meridional path whereas TCs that do not produce high surge are weaker and/or recurve out to sea. The highest surge for TCs that produce surge exceeding 1-yr return levels also generally occurs when the TC is located to the southwest of each location, allowing for onshore winds to push water towards the coastline. For locations that are further south, the picture is more complicated as TCs approach from different*

*directions. For these southern locations, there seems to be greater dependence on TC intensity than on TC path angle. While a majority of the TCs that produce surge exceeding the 1-yr return levels at Charleston, SC, Fort Pulaski, GA, and Fernandina Beach, FL generally move in a north-westward direction over Florida, nearly all of them have an average intensity around the time of surge maximum of 980 hPa or less." [L327 – 345]*

Introduction – specific comments

Nowhere in the introduction is there any mention of the TC characteristics that will be under consideration in this manuscript. Please add this description.

**We have specifically mentioned the characteristics we analyze as follows:**

> *"While these factors are important to surge, our focus will be on characteristics related to TCs, including the TC proximity to a tide gauge, TC intensity, measured through its mean sea-level pressure (MSLP), TC path angle, and TC propagation speed, all of which can be ascertained from historical cyclone track information." [L73 – 76]*

The introduction (more specifically, the second paragraph, lines 33 – 44) makes it seem like there will also be a focus on ETCs. Please make it explicitly clear you will solely focus on the TCs.

**At the end of this paragraph, we have added the following to clarify this:**

> *"Thus, even though TCs occur much less frequently than ETCs along the US east coast (e.g., Booth et al., 2016), individual TCs can cause more damage as they often are associated with more moisture and stronger winds than ETCs. Therefore, it is the focus of this research to understand how differences in certain characteristics of TCs relate to storm surge." [L48 – 51]*

Line 24-25: exposure is not the same as vulnerability

**This is an important differentiation. We rephrased this sentence as:**

> *"Population increases and development without adequate planning for hazards in coastal regions has led to an increase in exposure and vulnerability in low-lying areas (e.g., Strauss et al., 2012; Hallegatte et al., 2013)." [L26 – 27]*

Line 29-30: The amount of destruction is also influenced by changes in exposure and vulnerability

**This comment has been incorporated into the opening sentence noted in the previous comment.**

Line 30-31: What do you mean with this sentence?

**We have removed this sentence from the introduction in order to be more concise.**

Line 35: Could you please elaborate on what these differences exactly are?

**This sentence was updated to:**

> *"For ETCs, different atmospheric circulation patterns can produce large surge, with the highest median surge occurring with a slow-moving ETC in conjunction with an anticyclone located to its north (Catalano and Broccoli, 2018)." [L35 – 37]*

Line 36 – 38: Please explain to the reader why these cities have less TC-related storm surge extremes (along the lines of ocean waters are colder + more wind shear, so less favorable for TCs), that will also help the reader understand why this Boston example is noteworthy.

**We have added in this information to address this:**

> *"This is because at higher latitudes, TCs encounter environmental conditions that do not promote the sustainability of TCs, including cooler sea surface temperatures and increased wind shear associated with the jet stream, particularly later in the Atlantic hurricane season." [L39 – 41]*

Line 38 – 44: Please fill in the gaps that are left in this paragraph: 1) what are the differences in storm dynamics? 2) What are the different characteristics of the flood exceedance curves? 3) What exactly is the frequency of TCs compared to ETCs? 4) How can they cause more damage?

**This paragraph has been updated to include this information as follows:**

> *"Although both TCs and ETCs can generate surge, it is important to note that some of the energetics of the atmosphere differ for TCs and ETCs. While both TCs and ETCs are fundamentally low-pressure systems, TCs derive their energy through latent heat release over warm ocean waters, whereas ETCs gain their energy from the presence of air masses with different temperature and moisture characteristics (e.g., Jones et al., 2003; Yanase and Niino, 2015). Due to these differences in storm dynamics, flood exceedance curves for TCs and ETCs can exhibit different characteristics when considering long timescales (i.e., 100-yr events) as more extreme events are likely to be associated with TCs (Orton et al., 2016). Thus, even though TCs occur much less frequently than ETCs along the US east coast (e.g., Booth et al., 2016), individual TCs can cause more damage as they often are associated with more moisture and stronger winds than ETCs. Therefore, it is the focus of this research to understand how differences in certain characteristics of TCs relate to storm surge." [L43 – 51]*

Line 45 – 54: Could you please summarize this in a few sentences? Also: the term "noisy" is very vague. I also feel like the literature is very tailored towards US case studies and misses some other relevant studies (see my earlier comment)

**We have attempted to summarize these statements as well as include other relevant studies suggested earlier in the following:**

> *"Synthetic TC tracks along the Mid-Atlantic and the Northeast US have been heavily utilized to identify various relationships between surge and wind speed (Lin et al.,*

*2010), TC tracks (Garner et al., 2017), and landfall angle (Ramos-Valle et al., 2020). Additionally, Camelo et al. (2020) simulated 21 storms in the Gulf of Mexico and along the east coast of the US and found no individual TC characteristic correlates well with storm surge. The effect of the size of hurricanes on storm surge was found to be significant in the Gulf of Mexico (e.g., Irish et al., 2008; Needham and Keim, 2014). While comparing both observed and modeled surge heights, Bloemendaal et al. (2019) affirmed that surge height is influenced by the intensity and size of TCs in addition to coastal complexities and slope. Peng et al. (2006) examined the sensitivity of surge induced by both offshore and onshore winds to wind speed and direction. Needham and Keim (2014) empirically found that storm surge correlates better with TC winds pre-landfall as opposed to winds at landfall; Roberts et al. (2015) found a similar result for all storm types. Modeling work also suggests that with anthropogenic climate change, TCs will become stronger and peak intensity will occur at higher latitudes, and thus, changes to the intensity, frequency, and tracks of TCs are likely to impact storm surge (Knutson et al., 2020)."* **[L53 – 64]**

**Additionally, we have removed the use of the term "noisy" and include better descriptions of the data where applicable. Since our research focuses on surge along the east coast of the US, this literature is most relevant in describing the motivation for our research.**

Line 54 – 56: I strongly disagree with the wording here. The authors make it seem like they will overcome the regional scale, but they still perform a regional assessment (namely, the US East Coast).

**Our statement was to imply that not many studies have used historical observations and connected them to TC characteristics. Many of the studies cited here have adjusted characteristics of TC tracks synthetically or used model simulations as opposed to what we are doing in utilizing past observations, isolating the maximum surge produced by a TC and examining what characteristics those TCs exhibited at the time of the surge maximum. We have rephrased this statement to the following:**

*"While many studies have focused on utilizing synthetic tracks and models to better understand the relationship between storm surge and TCs, to our knowledge, no previous assessment has examined historical surge observations with a focus on surge variability relative to TC characteristics in addition to calculating storm surge exceedance probabilities based on TC characteristics. Therefore, we have designed an analysis to utilize past observations to determine the correlation between storm surge and TC characteristics as well as utilize those characteristics to determine the likelihood of surge exceeding some threshold at various locations along the eastern US."* **[L64 – 69]**

Line 59 – 60: Perhaps good to also mention coastal complexity here (Bloemendaal et al 2019)

**Thanks, we have added this study here.**

Methods – specific comments

Line 86 – 87: This is quite a bold statement to make, without any additional clarification. How big is the contribution of TC waves to total water levels?

**We have rephrased this sentence to account for the complexities associated with the wave setup and its contribution to surge:**

> *"While the wave setup is an important component to the water level (e.g., Phan et al., 2013, Marsooli and Lin, 2018), we neglect this component in our calculation of storm surge due to its overall complexities and its variations based on location and storm intensity. Additionally, the wave setup in the non-tidal residual is minimal because tide gauges are typically located in protected areas, such as harbors and bays."* **[L102 – 105]**

Table 1: Please round the pressure to one decimal place

**This change has been made.**

Line 104: What do you mean with wind intensity? Wind speed? What is the time reference for this wind speed? (1-min, 10-min, 3-sec?) Please also add units with every TC characteristic listed here.

**Wind intensity is the maximum sustained surface wind speed, as defined in the HURDAT2 database. We have added this information to the following sentence:**

> *"The TC variables we utilize are its location, central MSLP minimum (units: hPa), and maximum sustained surface wind speed, defined as the maximum 1-min average wind speed at 10 m (units: knots)."* **[L119 – 121]**

Line 105: Please state the exact dimensions of the "specified distance"

**We have clarified this and replaced it with the numerical value of 500 km.**

Line 123: Why are you solely testing for linear relationships?

**We expanded this to include exponential fits and now show both types of regression analyses.**

Line 130: Please explain to the reader what these results are

**We have added this information to the following sentence:**

> *"This choice of timing is motivated by the results of Needham and Keim (2014) who found storm surge best correlates with TC winds 18 h prior to landfall."* **[L156 – 157]**

Figure 2: Why are you differentiating between a radius of 250 km (in Figure 2) and 500 km (in the text)? Also, how did you derive the track angle? To me it seems like one of the green tracks for Charleston has a N-NNW angle upon approaching the landfall location, but it is listed as SW.

**Figure 2 has been updated to be for a radius of 500 km. The track angle calculation is described on L166 – 172. The calculated track angle is not relative to landfall location, it is relative to the time of the surge maximum, so depending on when the surge maximum occurs, the TC can be moving in various directions. Additionally, we average the track angle over the time period from 18 hours prior to the surge maximum to 6 hours post surge maximum. We have updated this paragraph to include these clarifications:**

> *"For the calculation of track path angle, we calculate the change in latitude and longitude between time steps separated by five time steps along the track of the TC. This method allows us to examine the change in the direction of the TC over a longer period of time as opposed to between consecutive time steps. The atan2d function in MATLAB is then utilized to find the TC path angle, as this function returns the four-quadrant inverse tangent. The track path angles range from 0° or 360° (eastward) to 90° (northward) to 180° (westward) to 270° (southward). Examples of TC tracks and their respective path angles for New York, NY and Charleston, SC are shown in figure 1. The TC path angles are not grouped relative to the site of the tide gauge, rather they are relative to the direction the TC is moving around the time of the surge maximum."* **[L166 – 172]**

Results – specific comments

To me, a 0.5-yr return level of 0.8m seems like a lot. This implies that (assuming the authors correctly identified individual storm surge events) Sandy Hook is affected by TC storm surges of this height on average twice per year! Could you please validate these results against other studies?

**Thanks for bringing this to our attention. We found in our calculation of return periods that we were using hourly data and the independence threshold was in hours not days, so this obviously skewed our results. We have now updated this for daily data and include an independence threshold of 2 days. After implementing this change, the return levels and periods make more sense.**

Please quantify the statistical significance and correlation throughout the results-section.

**We have added this information in to the results section where applicable.**

A lot of results aren't shown (indicated by "not shown" in the text). Could you please add these results to the supplementary materials, so that the reader can have a look at these results?

**We have included 5 supplemental figures.**

The continuous switching between a 500 km and 250 km radius is highly confusing – please re-read this section carefully and try to homogenize this usage of radii.

**The reasoning behind using different radii is to illustrate the importance of TC proximity with respect to surge, with closer (and stronger) TCs more frequently associated with higher surge. We have added this to the discussion:**

*"As the distance decreases from 500 km to 100 km, the percentage of TCs producing storm surge that exceeds the 1-yr return level increases. This would indicate that as a TC gets closer, the likelihood that it produces high surge is greater than if it were at a further distance."* **[L295 – 297]**

Conclusions

Line 329 – 333: Please check the work of Ramos-Valle et al (2020); they synthetically changed details of various storms in the Mid-Atlantic Bight.

**We have rephrased this sentence to the following:**

*"For the more northern sites, one might consider testing the hypothesis using numerical modelling, in which one could model a single TC and synthetically change details of the storm, as done previously by Lin et al. (2010), Garner et al. (2017), and Ramos-Valle et al. (2020). However, we want to emphasize that such an approach is very different from our work herein, because in the observational dataset it is not possible to ensure that only one characteristic of a TC varies while all others remain constant."* **[L390 – 394]**

---

## Referee Report (RR1)

Towey et al "Tropical cyclone storm surge probabilities for the east coast of the United States: A cyclone-based perspective"

I applaud the authors for the thorough revisions of their manuscript. The current manuscript has certainly improved compared to the previous version and I feel that it is almost ready to be accepted for publication. Please find below some minor comments:

Line 40: a temperature isn't warm or cold, but high or low

Line 122: You could also argue that a TC passing > 500 km from a site generally has a limited impact

Line 126: could you add the respective wind speed threshold on the Saffir-Simpson Hurricane Wind Scale here?

Line 142: I would leave out the mention of ETCs here, as this is confusing

Table 2: Strictly speaking is this table showing results; I would therefore move this to the Results-section (as you discuss the table there).

Line 208 - 209, Line 224-225: I thought you left out the extratropical cyclones? If this is the case, I would try to limit the mentioning of ETCs or ET transition, as these sentences seem to imply that ETCs are also being investigated (plus, I believe that the storm surges you are looking at are solely those of storms classified as TCs?)

Line 236: Though I understand you want to mention Sandy and its extraordinary track here, I would rephrase this mentioning of Hurricane Sandy, because the current write-up is confusing to the reader. I would suggest to say something along the lines that "Some readers might be familiar with the most infamous event in this region, Hurricane Sandy, which had a SE-NW track orientation that substantially differed from this general northeastern movement (Hall and Sobel 2013). However, as Sandy had underwent extratropical transition upon approaching New Jersey, this event is left out from our analysis as we solely focus on TCs."

Perhaps an interesting feature to help clarify your results for the direction is the orientation of the wind field? (which is counterclockwise in the NH). You do briefly mention this in the paragraph below Figure 6, but perhaps it's helpful for the reader to also be "reminded" of this in the discussion of Figure 5, where you do discuss the orientation of the track and its effect on storm surge heights.

Line 272 (but also other instances in the Results-section wherever appropriate): Could you please explain (from a physical point of view) what this negative correlation means?

Table 3: Could you please use an asterisk (*) instead of italics to indicate the statistically (not) significant results? I had a hard time finding the italicized numbers.

Line 292: I would leave out the mention of the hurricane season here, because technically speaking a TC can also occur outside of the hurricane season.

Lastly, following the Editor's suggestion (See "Editor decision" on 28 November 2021), I would suggest to add a paragraph/a few lines of discussion on how your research can be applied to other types of storms. How can one use your approach to study other types of storms and in other regions, and what kind of data is necessary to conduct such research?

---

## Author Response (AR2)

Response to the editor and reviewers:

Thank you for taking the time to review this manuscript once again and provide further feedback. We have addressed the concerns of reviewer #2, Dr. Bloemendaal, and believe these changes have further improved the clarity and impact of our results.

**Starting on the following page, we provide point-by-point responses to all questions and suggestions from Dr. Bloemendaal.**
- The reviewer comments are in black.
- **Our responses are indented and written in a bold blue.**
- **The line numbers in the revised manuscript that correspond to the changes made are added to the end of each comment as [L#-#].**

I applaud the authors for the thorough revisions of their manuscript. The current manuscript has certainly improved compared to the previous version and I feel that it is almost ready to be accepted for publication. Please find below some minor comments:

Line 40: a temperature isn't warm or cold, but high or low

**We have changed "cooler sea surface temperatures" to "lower sea surface temperatures." [L41]**

Line 122: You could also argue that a TC passing > 500 km from a site generally has a limited impact

**Good point, we have added this to the end of the sentence as follows:**
   *"Generally, a search radius beyond 500 km is too large when considering the spatial size of TCs (e.g., Booth et al., 2016) as TCs located beyond 500 km from a location will have limited impacts." [L128 – 129]*

Line 126: Could you add the respective wind speed threshold on the Saffir-Simpson Hurricane Wind Scale here?

**We have added this threshold to the end of the sentence as follows:**
   *"We consider all TCs in the HURDAT2 database that are categorized as a tropical storm or hurricane when the storm is within 500 km, meaning their maximum sustained wind speed is at least 34 knots." [L131 – 133]*

Line 142: I would leave out the mention of ETCs here, as this is confusing

**We agree and have removed this mention of ETCs.**

Table 2: Strictly speaking is this table showing results; I would therefore move this to the Results-section (as you discuss the table there).

**Agreed – we have now moved this table to the Results section. [L292]**

Line 208 – 209, Line 224-225: I thought you left out the extratropical cyclones? If this is the case, I would try to limit the mentioning of ETCs or ET transition, as these sentences seem to imply that ETCs are also being investigated (plus, I believe the storm surges you are looking at are solely those of storms classified as TCs?)

**The non-ET TCs are the primary focus of the paper, but in these instances we refer to supplemental figures that compare non-ET TCs and ET TCs for select sites. We have made the following clarifications:**

> *"Since we focus only on TCs that are considered purely tropical, i.e., they do not undergo ET (non-ET TCs), in this analysis, we include a supplemental figure to compare the relationship between surge and TC proximity for non-ET TCs against ET-TCs for the six most northern sites, which have at least 40% of their TCs undergo ET. When examining storm surge as a function of distance for ET TCs, the fit worsens compared to that for non-ET TCs for these six sites (S2)."* [L211 – 215]

> *"This analysis of ET TCs highlights the complexities associated with the change in storm dynamics as a TC transitions into an ETC and is why we exclude these TCs from our primary analysis."* [L230 – 232]

Line 236: Though I understand you want to mention Sandy and its extraordinary track here, I would rephrase this mentioning of Hurricane Sandy, because the current write-up is confusing to the reader. I would suggest to say something along the lines that "Some readers might be familiar with the most infamous event in this region, Hurricane Sandy, which a SE-NW track orientation that substantially differed from this general northeastern movement (Hall and Sobel 2013). However, as Sandy had underwent extratropical transition upon approaching New Jersey, this event is left out from our analysis as we solely focus on TCs."

Perhaps an interesting feature to help clarify your results for the direction is the orientation of the wind field? (which is counterclockwise in the NH). You do briefly mention this in the paragraph below Figure 6, but perhaps it's helpful for the reader to also be "reminded" of this in the discussion of Figure 5, where you do discuss the orientation of the track and its effect on storm surge heights.

**Thank you for this comment. We have adjusted the discussion of Figure 5 to say the following:**

> *"For locations north of Cape May, NJ, the largest storm surge events tend to occur as TCs move toward the northeast, in which onshore winds associated with the counterclockwise flow around the TC would push water toward the coastline. Hurricane Sandy, one of the most infamous events to occur in this region, was a unique system in part due to its southeasterly track toward the New Jersey coastline, which varied considerably from the general track direction toward the northeast that is more commonly observed in this region (Hall and Sobel, 2013). Since our primary focus in this analysis is on TCs that do not undergo ET, Sandy is not included in this analysis as it underwent ET upon approaching New Jersey."* [L242 – 248]

Line 272 (but also other instances in the Results-section wherever appropriate): Could you please explain (from a physical point of view) what this negative correlation means?

**We have added this clarification in the following spots:**

*"We also examined the influence of propagation speed (S4) and found a negligible correlation with storm surge, suggesting that the magnitude of storm surge does not have a clear relationship with propagation speed."* **[L258 – 259]**

*"This negative correlation suggests that as TC proximity to a location decreases, the magnitude of the storm surge increasing, highlighting the importance of TCs that are close to a location."* **[L285 – 286]**

Table 3: Could you please use an asterisk (*) instead of italics to indicate the statistically (not) significant results? I had a hard time finding the italicized numbers.

**This is now Table 2 due to moving the original Table 2 to the place of Table 3. We have added an asterisk to replace the use of italics in this table – thank you for the suggestion.**

Line 292: I would leave out the mention of the hurricane season here, because technically speaking a TC can also occur outside of the hurricane season.

**This is true. We have removed this mention of hurricane season.**

Lastly, following the Editor's suggestion (See "Editor decision" on 28 November 2021), I would suggest to add a paragraph/a few lines of discussion on how your research can be applied to other types of storms. How can one use your approach to study other type of storms and in other regions, and what kind of data is necessary to conduct such research?

**Thank you for this suggestion. We did briefly get to this at the very end of the last paragraph in the Conclusion section, but we have further expanded to the following:**
*"This type of analysis, while limited to the east coast of the US, can be applied to any region with a record of observations associated with any type of hazard to be used in conjunction with any cyclone dataset. This cyclone-hazard association algorithm has been applied to associating precipitation and streamflow events with both TCs and ETCs in the Catskill Mountains of New York state (Towey et al., 2018). In this instance, ETC tracks were identified by applying a Lagrangian tracking algorithm (Bauer et al., 2016), which follows centers of low sea-level pressure, to reanalysis data.*

*Similar to our analysis presented herein, Lionello et al. (2019) linked sea level anomalies to the intensity and position of cyclones in the Mediterranean Sea through the use of a cyclone tracking algorithm. Given any observational dataset for a location and a cyclone tracking algorithm, this type of analysis can be utilized to conduct similar research for any region."* **[L422 – 429]**